# Egr2 induction in spiny projection neurons of the ventrolateral striatum contributes to cocaine place preference in mice

Diptendu Mukherjee[1,2†], Ben Jerry Gonzales[1,2†], Reut Ashwal-Fluss[1], Hagit Turm[1,2], Maya Groysman[1], Ami Citri[1,2,3]*

[1]The Edmond and Lily Safra Center for Brain Sciences, Jerusalem, Israel; [2]Institute of Life Sciences, The Hebrew University of Jerusalem, Jerusalem, Israel; [3]Program in Child and Brain Development, Canadian Institute for Advanced Research, MaRS Centre, Toronto, Canada

**Abstract** Drug addiction develops due to brain-wide plasticity within neuronal ensembles, mediated by dynamic gene expression. Though the most common approach to identify such ensembles relies on immediate early gene expression, little is known of how the activity of these genes is linked to modified behavior observed following repeated drug exposure. To address this gap, we present a broad-to-specific approach, beginning with a comprehensive investigation of brain-wide cocaine-driven gene expression, through the description of dynamic spatial patterns of gene induction in subregions of the striatum, and finally address functionality of region-specific gene induction in the development of cocaine preference. Our findings reveal differential cell-type specific dynamic transcriptional recruitment patterns within two subdomains of the dorsal striatum following repeated cocaine exposure. Furthermore, we demonstrate that induction of the IEG *Egr2* in the ventrolateral striatum, as well as the cells within which it is expressed, are required for the development of cocaine seeking.

*For correspondence:
ami.citri@mail.huji.ac.il

†These authors contributed equally to this work

Competing interests: The authors declare that no competing interests exist.

## Introduction

Psychostimulant addiction is characterized by life-long behavioral abnormalities, driven by circuit-specific modulation of gene expression (*Nestler, 2014*; *Nestler and Lüscher, 2019*; *Salery et al., 2020*; *Steiner, 2016*). Induction of immediate-early gene (IEG) transcription in the nucleus accumbens (NAc) and dorsal striatum (DS) are hallmarks of psychostimulant exposure (*Berke et al., 1998*; *Caprioli et al., 2017*; *Chandra and Lobo, 2017*; *Gao et al., 2017b*; *Gerfen, 2000*; *Gonzales et al., 2020*; *Guez-Barber et al., 2011*; *Hope et al., 1994*; *Moratalla et al., 1996*; *Mukherjee et al., 2018*; *Nestler et al., 1993*; *Nestler, 2001*; *Nestler and Aghajanian, 1997*; *Piechota et al., 2010*; *Turm et al., 2014*). As such, IEG induction has been utilized to support the identification of functional neuronal assemblies mediating the development of cocaine-elicited behaviors ('cocaine ensembles'; *Bobadilla et al., 2020*; *Cruz et al., 2013*). Within these striatal structures, the principal neuronal type is the spiny projection neuron (SPN), which is comprised of two competing subtypes, defined by their differential expression of dopamine receptors. Expression of the D1R dopamine receptor is found on direct-pathway neurons, responsible for action selection by promoting behavioral responses, while D2R-expressing indirect pathway neurons are responsible for action selection through behavioral inhibition (*Kreitzer and Malenka, 2008*; *Lipton et al., 2019*). In the striatum, the cellular composition of cocaine ensembles varies by domain: Fos-expressing cocaine ensembles in the NAc are enriched for D1R expression (*Koya et al., 2009*), while in the DS, IEG expression and psychostimulant-responsive ensembles are spatially segregated to the medial striatum (MS) and ventrolateral striatum (VLS), encompassing both D1R$^+$ and D2R$^+$ neurons in the MS, and enriched for

**eLife digest** The human brain is ever changing, constantly rewiring itself in response to new experiences, knowledge or information from the environment. Addictive drugs such as cocaine can hijack the genetic mechanisms responsible for this plasticity, creating dangerous, obsessive drug-seeking and consuming behaviors.

Cocaine-induced plasticity is difficult to apprehend, however, as brain regions or even cell populations can react differently to the compound. For instance, sub-regions in the striatum – the brain area that responds to rewards and helps to plan movement – show distinct responses during progressive exposure to cocaine. And while researchers know that the drug immediately changes how neurons switch certain genes on and off, it is still unclear how these genetic modifications later affect behavior.

Mukherjee, Gonzales et al. explored these questions at different scales, first focusing on how progressive cocaine exposure changed the way various gene programs were activated across the entire brain. This revealed that programs in the striatum were the most affected by the drug.

Examining this region more closely showed that cocaine switches on genes in specific 'spiny projection' neuron populations, depending on where these cells are located and the drug history of the mouse. Finally, Mukherjee, Gonzales et al. used genetically modified mice to piece together cocaine exposure, genetic changes and modifications in behavior. These experiments revealed that the drive to seek cocaine depended on activation of the *Egr2* gene in populations of spiny projection neurons in a specific sub-region of the striatum. The gene, which codes for a protein that regulates how genes are switched on and off, was itself strongly activated by cocaine intake.

Cocaine addiction can have devastating consequences for individuals. Grasping how this drug alters the brain could pave the way for new treatments, while also providing information on the basic mechanisms underlying brain plasticity.

D1R expression in the VLS (*Caprioli et al., 2017*; *Cruz et al., 2015*; *Gonzales et al., 2020*; *Li et al., 2015*; *Rubio et al., 2015*; *Steiner and Gerfen, 1993*). The VLS subregion partially overlaps with a lateral striatum segment enriched for *Gpr155* expression, defined in recent molecular striatal subdivisions (*Märtin et al., 2019*; *Ortiz et al., 2020*).

Depending on the history of prior cocaine exposure, a unique pattern of IEG induction is observed across brain structures (*Mukherjee et al., 2018*). This transcriptional code was characterized addressing a handful of transcripts within bulk tissue, warranting a comprehensive study of the induced gene expression programs across key structures of the reward circuitry. Here we comprehensively describe gene programs in progressive stages of cocaine experience across multiple brain structures, analyze the spatial and cell-type-specific patterns of IEG expression within prominently recruited brain regions, and functionally link induced gene expression to the development of cocaine preference.

Taking an unbiased approach to the identification of the cellular and molecular modifications underlying the development of cocaine-elicited behaviors, we analyzed dynamics of cocaine-induced transcription across five structures of the reward circuitry. Of these, the most prominently induced gene programs were in the DS. Addressing the spatial segregation of these transcriptional programs within the DS (studying 759,551 individual cells by multiplexed single-molecule fluorescence in-situ hybridization), we investigated the dynamics of cell-specific recruitment within the two striatal sub-domains engaged by cocaine, the MS and VLS. While both D1R$^+$ and D2R$^+$ neurons in the MS were engaged transcriptionally throughout the development of cocaine sensitization, the recruitment of D1R$^+$ neurons in the VLS fluctuated depending on the history of cocaine exposure. The IEG *Egr2*, which we find to be the most robustly induced following cocaine experience, serves as a prominent marker for these VLS ensembles. We therefore addressed the function of VLS Egr2$^+$ ensembles, as well the role of VLS expressed Egr2-transcriptional complexes, in the development of cocaine seeking. Our results identify the VLS as a hub of dynamic transcriptional recruitment by cocaine and define a role for Egr2-dependent transcriptional regulation in VLS D1R$^+$ neurons in the development of cocaine seeking.

## Results

### Characterization of transcriptional dynamics in the reward circuitry during the development of behavioral sensitization to cocaine

In order to characterize brain-wide gene expression programs corresponding to the development of psychostimulant sensitization, we exposed mice to cocaine (20 mg/kg, i.p.) acutely, or repeatedly (five daily exposures), as well as to a cocaine challenge (acute exposure following 21 days of abstinence from repeated exposure to cocaine) (*Figure 1A*). We then profiled transcription (applying 3′-RNA-seq) within key brain structures of the reward circuitry (limbic cortex = LCtx, nucleus accumbens = NAc, dorsal striatum = DS, amygdala = Amy, lateral hypothalamus = LH; see *Figure 1—figure supplement 1* for the delineation of brain tissue dissected; *Supplementary file 1* and *Figure 1—figure supplement 2* for a description of the samples sequenced) at 0 (not exposed to cocaine on day of sample collection), 1, 2 or 4 hr post-cocaine exposure (*Figure 1A*). Mice exhibited increased locomotion upon acute exposure to cocaine, further increasing following repeated exposure and maintained after abstinence and challenge re-exposure, typical of locomotor sensitization to this intermediate cocaine dose (*Figure 1B*, $F_{8,312} = 178.9$, $p<0.0001$, ANOVA).

### Repeated cocaine administration and abstinence induce prominent transcriptional shifts across multiple brain regions

Experience impacts gene transcription at multiple timescales (*Clayton et al., 2020*; *Mukherjee et al., 2018*; *Nestler and Lüscher, 2019*; *Rittschof and Hughes, 2018*; *Sinha et al., 2020*; *Yap and Greenberg, 2018*). Whereas the expression of inducible genes peak and decay on a time scale of minutes-to-hours following stimulation, baseline shifts in brain-wide gene expression programs are also observed following more prolonged periods (days to weeks) (*Clayton et al., 2020*), presumably implementing, supporting, and maintaining the modified behavioral output (*Sinha et al., 2020*). We initially focused on baseline shifts in gene expression, comparing naïve mice (never exposed to cocaine) to mice exposed repeatedly to cocaine, as well as to mice following 21 days of abstinence from repeated cocaine exposure (*Figure 1C*; *Figure 1—figure supplement 3A*; refer to *Supplementary file 2* for list of differentially expressed genes and normalized counts). Differentially expressed genes (DEGs) included both upregulated and downregulated genes across all brain regions analyzed, with prolonged abstinence driving the most extreme shifts in expression (*Figure 1—figure supplement 3B,C*). While gene-expression shifts following repeated exposure to cocaine were prominent in the DS, abstinence-induced changes were more prominent in the NAc and LCtx (*Figure 1—figure supplement 3C*). KEGG analysis demonstrated that DEGs were enriched for synaptic genes and disease pathways (*Figure 1—figure supplement 3D*). To provide insight into the cellular mechanisms affected by repeated drug exposure and abstinence, we implemented Gene Ontology (GO term) enrichment analysis (*Figure 1C*, *Figure 1—figure supplement 4*, see *Supplementary file 3* for definition of clusters and DEGs included within them). Gene clusters associated with synaptic plasticity, myelin, and proteostasis demonstrated shifts in expression across multiple brain structures, whereas a cluster of genes associated with structural plasticity appeared more specific to striatal structures (DS and NAc). Noteworthy gene clusters that displayed modified expression were involved in cell–cell communication; glutamate-induced plasticity; synaptic vesicle formation, transport, and fusion; actin filament components; and projection morphogenesis. Notably, the expression of protein folding genes was coordinately upregulated across structures, while myelin components were coordinately downregulated (*Figure 1—figure supplement 4*). These results exemplify the dramatic shifts of transcription occurring in the brain in response to repeated cocaine exposure, potentially supporting maladaptive neuroplasticity driving drug addiction (*Bannon et al., 2014*; *Lull et al., 2008*).

### Transcriptional profiling illustrates dynamic recruitment of the striatum during the development of behavioral sensitization to cocaine

Inducible transcription supports the development of plastic changes following psychostimulant experience (*Alberini, 2009*; *Han et al., 2019*; *McClung and Nestler, 2008*; *Nestler and Lüscher, 2019*). We therefore assessed the inducible transcription response at 1, 2, or 4 hr following acute, repeated, or challenge cocaine exposure, observing robust IEG induction across all brain structures studied

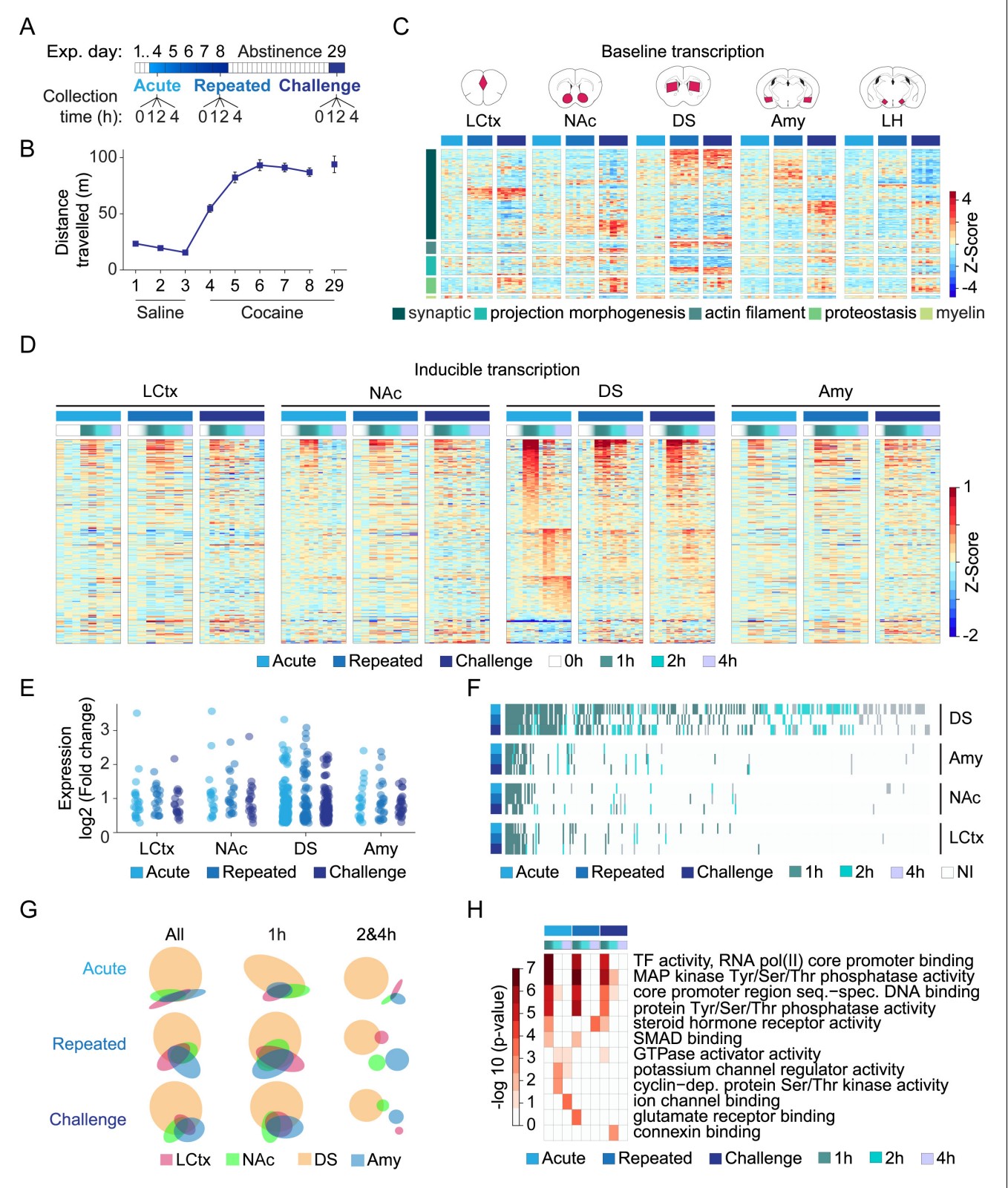

**Figure 1.** Transcriptional profiling resolves the dynamics of cocaine-induced gene expression within major nodes of the reward circuitry. (**A**) Scheme describing the cocaine sensitization paradigm and time points (0, 1, 2, 4 hr) at which samples were obtained for analysis of gene expression following acute (0 = cocaine naïve); repeated (fifth exposure to cocaine; 0 = 24 hr following fourth exposure); and challenge exposures (acute exposure following 21 days of abstinence from repeated exposure; 0 = abstinent mice). (**B**) Locomotor sensitization to cocaine (20 mg/kg i.p.; days 1–3 n = 58; days 4

*Figure 1 continued on next page*

*Figure 1 continued*

n = 51; day 8 n = 30; day 29 n = 15) of mice included in this study. (C) Baseline shifts in expression of genes associated with categories of neuroplasticity following repeated cocaine exposure and abstinence (see *Figure 1—figure supplements 1* and *2* for description of sectioned regions and RNA-seq QC). Heatmap depicting fold change of differentially expressed genes (normalized to cocaine naive samples and Z-scored per gene), with rows corresponding to individual genes, clustered according to annotation of biological function on Gene Ontology (p<0.05 FDR corrected). Columns correspond to individual mice – naïve (=azure); repeated (=blue); challenge (=navy) cocaine; n = 6–8 samples in each group across brain structures (LCtx = limbic cortex, NAc = nucleus accumbens, DS = dorsal striatum, Amy = amygdala and LH = lateral hypothalamus). Genes were selected from analysis of a subset of samples which were sequenced together (*Figure 1—figure supplement 3A*) and plotted here across all available samples (for gene identity, see *Figure 1—figure supplement 4*). (D) Heatmaps depicting expression of inducible genes. Data was normalized to 0 hr of relevant cocaine experience, log-transformed, and clustered by peak expression (selected by FC > 1.2 and FDR corrected p<0.05, linear model followed by LRT, see Materials and methods). Columns correspond to individual mice (0, 1, 2, 4 hr following acute, repeated vs challenge cocaine; see adjacent key for color coding) across LCtx, NAc, DS and Amy. n = 2–4 samples for individual time points of a cocaine experience within a brain nucleus. (E) Dot plots represent the peak induction magnitude of genes induced in the LCtx, NAc, DS, and Amy following acute, repeated, and challenge cocaine. (F) Heatmap addressing the conservation of gene identity and peak induction time. Induced genes are color coded by their time point of peak induction (NI = not induced). (G) Venn diagrams represent overlap of the genes induced in each brain nuclei following different cocaine experiences (all: 1 and 2 and 4 hr; early: 1 hr; late: 2 hr and 4 hr time points). (H) DEGs induced within the DS are enriched for GO terms associated with signaling and transcription at 1 hr, diversifying to regulators of cellular function and plasticity at later times. Heatmap represents significantly enriched GO terms (p < 0.05, Bonferoni corrected), graded according to p-value.

The online version of this article includes the following figure supplement(s) for figure 1:

**Figure supplement 1.** Boundaries of dissected brain structures.

**Figure supplement 2.** Quality control analysis of RNA-seq experiments.

**Figure supplement 3.** Repeated cocaine administration and abstinence induce baseline shifts in gene transcription.

**Figure supplement 4.** Repeated cocaine exposure and abstinence alters the expression of gene clusters related to neuroplasticity.

(*Figure 1D*). The largest number of induced genes, as well as the highest fold induction levels, were found in the DS (*Figure 1D,E*; refer to *Supplementary file 4* for the identities of genes induced in each structure and cocaine condition).

To what extent do the transcription programs induced in the different structures share common attributes? To query the overlap in the identity of genes induced and their temporal induction patterns following the different schedules of cocaine exposure, we graphed the induced genes, color coding them according to their time of peak induction (1, 2, or 4 hr following cocaine) (*Figure 1F*). Thus, for example, if a gene was commonly induced across structures with a peak at 1 hr across cocaine regimens, this would be evident as a contiguous vertical green line. This graph reveals aspects of the logic of these inducible transcription programs, whereby (1) genes induced following the different cocaine schedules largely maintain the same temporal structure, i.e., if the peak induction of a given gene was observed at a defined time point in one program, its peak induction time was maintained across other programs; (2) following repeated cocaine exposure, we observe a substantial dampening of the transcriptional response in the DS, which recovers following cocaine challenge, recapitulating a significant proportion of the acute cocaine gene program; (3) all gene programs largely represent subcomponents of the program induced by acute cocaine in the DS. We further visualized the overlap in the identity of genes induced in the different structures using Venn diagrams (*Figure 1G*), illustrating that the overlap stems principally from the immediate component of the transcriptional program (peaking at 1 hr following cocaine), while transcripts induced at 2 or 4 hr following cocaine diverged between structures. Focusing on the most robust programs, induced in the DS, we found that gene clusters enriched at the 1 hr time points are related primarily to transcriptional regulation and synapse-to-nucleus signal transduction, while clusters related to modification of neural morphology and function were enriched at later time points (*Figure 1H*; refer to *Supplementary file 5*). Taken together, these results highlight robust transcriptional adaptations in the DS, positioning it as a major hub of cocaine-induced plasticity. Furthermore, our results illustrate the utilization of a conserved set of genes during the early wave of transcription following experience, followed by divergence of subsequent transcription, possibly to support region-specific mechanisms of plasticity (*Hrvatin et al., 2018*; *Walker et al., 2018*).

## IEG induction in subdomains of the DS is influenced by the history of cocaine exposure

Our observation of dynamic transcriptional responses to repeated cocaine exposure in the DS (*Figure 1*) motivated us to address the cellular and spatial distribution of this transcriptional plasticity. Recently, using single-molecule fluorescence in-situ hybridization (smFISH), we reported region-specific rules governing the recruitment of striatal assemblies following a single acute exposure to cocaine (*Gonzales et al., 2020*). We now revisited this spatial analysis, applying smFISH to expand the investigation of the striatal distribution of the IEGs *Arc*, *Egr2*, *Fos*, and *Nr4a1* throughout the development of cocaine sensitization (*Figure 2*; *Figure 2—figure supplements 1* and *2*; results from *Gonzales et al., 2020* serve as a reference for the effects of acute cocaine exposure).

Addressing an overview of induced expression of these IEGs, we observed robust induction of *Arc*, *Egr2*, *Nr4a1*, and *Fos* following acute cocaine exposure, which was dampened following repeated exposure to cocaine and reinstated following a challenge dose of cocaine, in-line with the results described in *Figure 1* (*Figure 2A–C*). To visualize the subdomains defined by IEG expressing cells, we applied 2D kernel density estimation on striatal sections following repeated and challenge cocaine and compared resulting patterns to those previously described following acute cocaine exposure (*Gonzales et al., 2020*). The prominent recruitment of IEG expression in the VLS observed following acute cocaine exposure was dampened drastically after repeated cocaine exposure, and re-emerged upon cocaine challenge. In contrast to the findings in the VLS, dampening of IEG induction in the MS, while evident, was more modest (*Figure 2D*). These results are quantified in *Figure 2E,F*. In the VLS, the fraction of robustly expressing cells of *Egr2* increased to 46 ± 10% after acute cocaine, decreased to 21 ± 4% following repeated cocaine, and subsequently increased to 40 ± 11% upon cocaine challenge. Similar dynamics were observed for *Fos*, where the fractions of suprathreshold cells were observed to be 37 ± 10%, 21 ± 4%, and 34 ± 7% following acute, repeated, and challenge cocaine, respectively. In contrast, in the MS, the fraction of cells expressing *Egr2* and *Fos* increased to 42 ± 8% and 40 ± 7% after acute cocaine, modestly decreased to 33 ± 3% and 35 ± 3% after repeated cocaine, and regained elevated induction of 43 ± 4% and 40 ± 4% following challenge cocaine (*Figure 2E* [mean ± SD]; *Egr2* VLS $F_{2,66}$ = 21.4, p<0.0001; *Fos* VLS $F_{2,30}$ = 4.9964, p=0.01; *Egr2* MS $F_{2,66}$ = 6.4, p=0.002; *Fos* MS $F_{2,30}$ = 3.1, p=0.06; ANOVA followed by Tukey's test; for detailed statistics refer to *Supplementary file 6*). With reference to expression levels, acute, repeated, and challenge cocaine-mediated puncta/cell expression in the VLS was observed to be 11.9 ± 3.8, 3.9 ± 0.8, 9.5 ± 3.4 for *Egr2* and 6.8 ± 2.3, 3.5 ± 0.5, 5.9 ± 1.6 for *Fos*, respectively. Comparing these to the MS, the expression levels were observed to be 9.2 ± 2.2, 6.5 ± 1.0, and 9.5 ± 1.2 for *Egr2* and 7.4 ± 1.5, 6 ± 0.8, and 7.5 ± 1.3 for *Fos* after acute, repeated, and challenge, respectively (*Figure 2F*) (mean ± SD; *Egr2* VLS $F_{2,66}$ = 21.7, p<0.0001; *Fos* VLS $F_{2,30}$ = 4.9, p=0.01; *Egr2* MS $F_{2,66}$ = 9.01, p=0.0003; *Fos* MS $F_{2,30}$ = 3.4, p=0.04; ANOVA followed by Tukey's test; for detailed statistics refer to *Supplementary file 6*). A similar trend was evident for the expression of *Arc* and *Nr4a1* in the VLS vs. the MS (*Figure 2—figure supplement 1A,B*). Notably, the expression of different IEGs was highly correlated within individual cells, defining overlapping populations of neurons responsive to the cocaine experiences studied. Once recruited by cocaine, neurons committed to co-expression of multiple IEGs to virtually identical levels (*Figure 2—figure supplement 2*; for detailed statistics, see *Supplementary file 6*). These data demonstrate the coherent co-expression of multiple IEGs within striatal assemblies during the development of behavioral sensitization to cocaine, likely to support mechanisms of long-term plasticity within these ensembles. In sum, the history of cocaine experience is reflected in the differential transcriptional recruitment of striatal subdomains, dampening drastically in the VLS following repeated exposure.

## The IEG response is selectively dampened in VLS *Drd1*+ SPNs following repeated cocaine

Striatal *Drd1*+-neurons are implicated in promoting actions, while *Drd2*+-neurons are implicated in the tempering and refinement of action selection (*Bariselli et al., 2019*). Differential IEG induction in *Drd1*+ vs *Drd2*+ expressing SPN ensembles is expected to shed light on the relative contribution of plasticity within each cell type to the development of cocaine behaviors. We have previously reported that acute exposure to cocaine induces *Egr2* expression in both *Drd1*+ and *Drd2*+ neurons in the MS, while more selectively inducing *Egr2* expression in *Drd1*+-neurons in the VLS

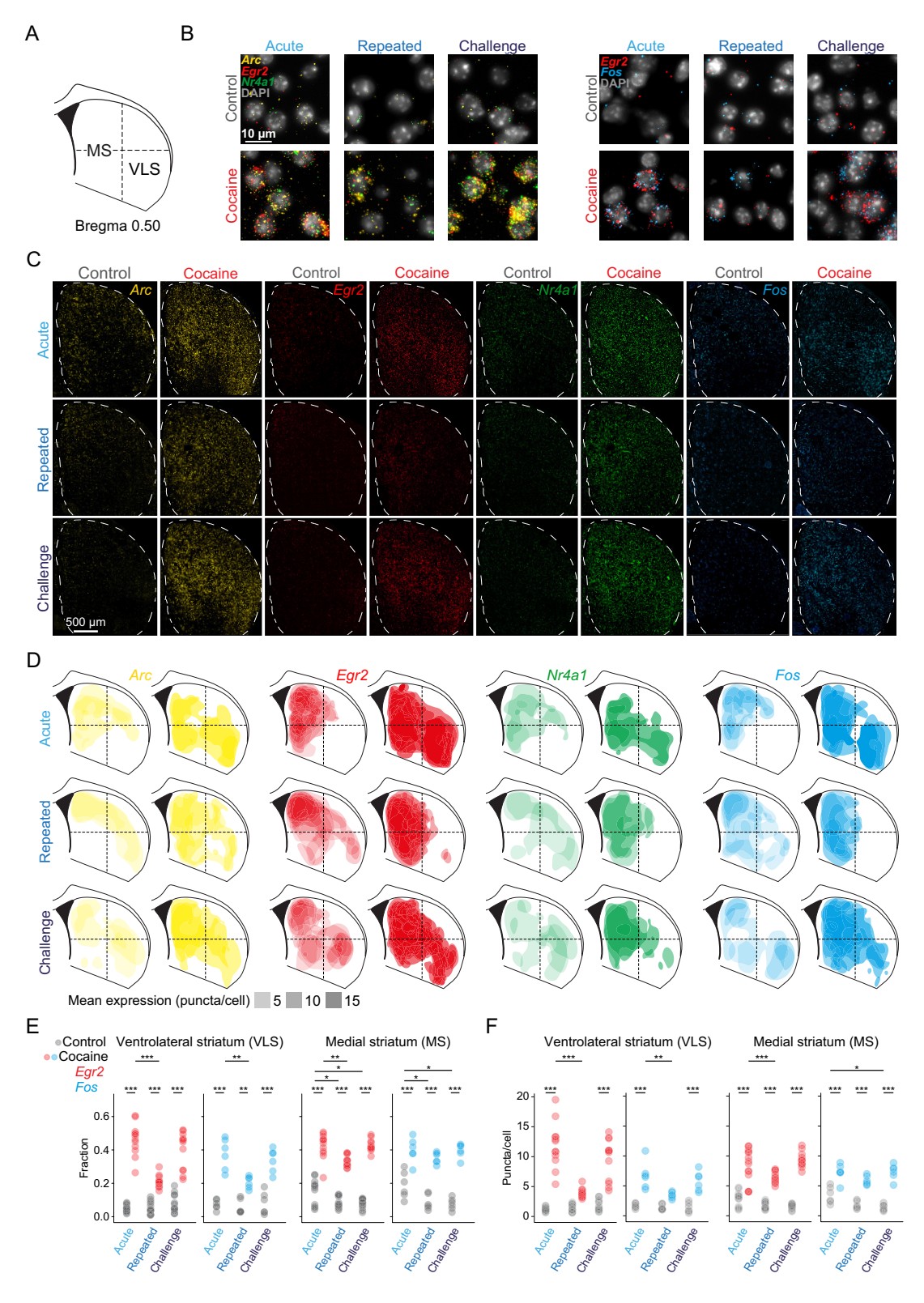

**Figure 2.** Dynamic IEG induction in subregions of the striatum accompany the development of cocaine sensitization. (**A**) Scheme of a coronal section of the dorsal striatum (DS) (+0.52 ± 0.1 mm from Bregma) corresponding to the region assayed by multicolor smFISH for cocaine-induced IEG expression. (**B**) Representative images of multicolor smFISH analysis of *Arc*, *Egr2*, *Nr4a1*, and *Fos* expression following acute, repeated and challenge cocaine exposures (40× magnification). (**C**) Spatial IEG expression patterns in the DS. Representative images of multicolor smFISH analysis of *Arc*, *Egr2*, *Nr4a1*,

*Figure 2 continued on next page*

*Figure 2 continued*

and *Fos* expression. (D) Cocaine experiences induce distinct spatial patterns of IEG expression. Two-dimensional kernel density estimation was used to demarcate the regions with maximal density of high expressing cells for each IEG. Color code for probes: *Arc* – yellow, *Egr2* – red, *Nr4a1* – green, *Fos* – blue. The opacity of the demarcated areas corresponds to the mean puncta/cell expression. (E, F) Dot plots depicting the proportion of cells suprathreshold for *Egr2*+ and *Fos*+ (fraction; E), as well as cellular expression (puncta/cell; F) of *Egr2*+ and *Fos*+ in the ventrolateral (VLS) and medial (MS) striatum following acute, repeated, and challenge cocaine. *p<0.05, **p<0.005, ***p<0.0001, two-way ANOVA with post hoc Tukey's test. Refer to *Supplementary file 7* for cell numbers. See *Figure 2—figure supplement 1* for corresponding analysis of *Arc* and *Nr4a1*. See *Figure 2—figure supplement 2* for correlation in expression of *Egr2*, *Arc*, and *Nr4a1*, as well as *Egr2* and *Fos*. Images relating to acute cocaine (in B, C, and D) were replicated from *Gonzales et al., 2020*, with permission.

The online version of this article includes the following figure supplement(s) for figure 2:

**Figure supplement 1.** Cocaine dynamically modulates cellular IEG expression in the VLS and MS.

**Figure supplement 2.** Coherence of cocaine-induced IEG expression.

(*Gonzales et al., 2020*). Extending this analysis to repeated and challenge cocaine exposures and with additional IEGs, we observed robust dampening of the induction of *Egr2* and *Fos* in VLS *Drd1*+ neurons following repeated exposure to cocaine, while upon cocaine challenge, prominent induction was again evident, especially in *Drd1* SPNs. (*Figure 3A–C*, *Figure 3—figure supplement 1*). In contrast, in the MS, subtle dampening was observed and *Egr2* and *Fos* expression maintained consistent correlation to *Drd1* and *Drd2* expression throughout acute, repeated, and challenge cocaine exposures (*Figure 3C*, *Figure 3—figure supplement 1*; for reference of *Drd1* and *Drd2* levels in MS and VLS, see *Figure 3—figure supplement 2*, *Supplementary file 6* for statistics). Thus, the observed attenuated transcriptional recruitment in the DS can be attributed to selective dampening of IEG induction, primarily within VLS *Drd1*+ neurons. This specialization in transcriptional plasticity likely underlies differential roles of the striatal subregions and cells within them in supporting behavioral modification induced by cocaine experience.

## Implication of VLS *Egr2* transcriptional activity in the development of cocaine-seeking behavior

The greater enrichment of *Egr2* induction within VLS neurons suggests a causal role for this neuronal population in supporting cocaine conditioned behaviors. To address the role of VLS *Egr2*+ neurons in cocaine seeking, we bilaterally injected Cre-dependent inhibitory hM4Di DREADD (VLS-Egr2^hM4Di), targeting the VLS of Egr2-Cre knock-in mice. In these mice, an *Egr2* allele is substituted for *Cre* (*Voiculescu et al., 2000*), supporting the expression of Cre recombinase in neurons expressing *Egr2*. DREADD hM4Di-mediated selective inhibition of the VLS *Egr2*-expressing neuronal ensembles was achieved by administration of clozapine-N-oxide (CNO) (*Atlan et al., 2018*; *Terem et al., 2020*). Control mice were either transduced with viruses expressing hM4Di, similar to the experimental group, and exposed to saline (*Figure 3E*) or transduced with viruses conditionally expressing mCherry and exposed to CNO (*Figure 3G*, for expression domains, see *Figure 3F*, *Figure 3—figure supplement 3A,B*).

In an initial experiment, we transduced two groups of mice with AAV-DIO-hM4Di to the VLS. Three weeks later, we ran a cocaine conditioned-place preference (CPP) experiment, in which mice were conditioned over three alternate days to a cocaine-associated context, while on the day following each conditioning session their side preference was tested (conditioning – days 2, 4, 6; tests – days 1, 3, 5, 7; *Figure 3D*, 'Design 1'). In control mice expressing hM4Di and exposed to saline prior to cocaine conditioning sessions, CPP developed following a single conditioning session and was reinforced following additional conditioning sessions (*Figure 3E*). The experimental group, which was exposed to CNO (5 mg/kg; i.p.) 30 min prior to cocaine conditioning, also displayed CPP following the initial exposure; however, in this group, the preference decayed with additional conditioning, such that following the third conditioning session, CPP in this group was significantly different from the control group (preference score saline vs. CNO Test3: p<0.05, t = 2, df = 8.6; one-tailed t-test; *Figure 3E*). We interpret these results as suggesting that the first cocaine conditioning session induced expression of Cre within VLS *Egr2*+ neurons, supporting the accumulation of functional hM4Di within these neurons to a CNO-responsive complement by the third conditioning session, resulting in diminished conditioned-place preference.

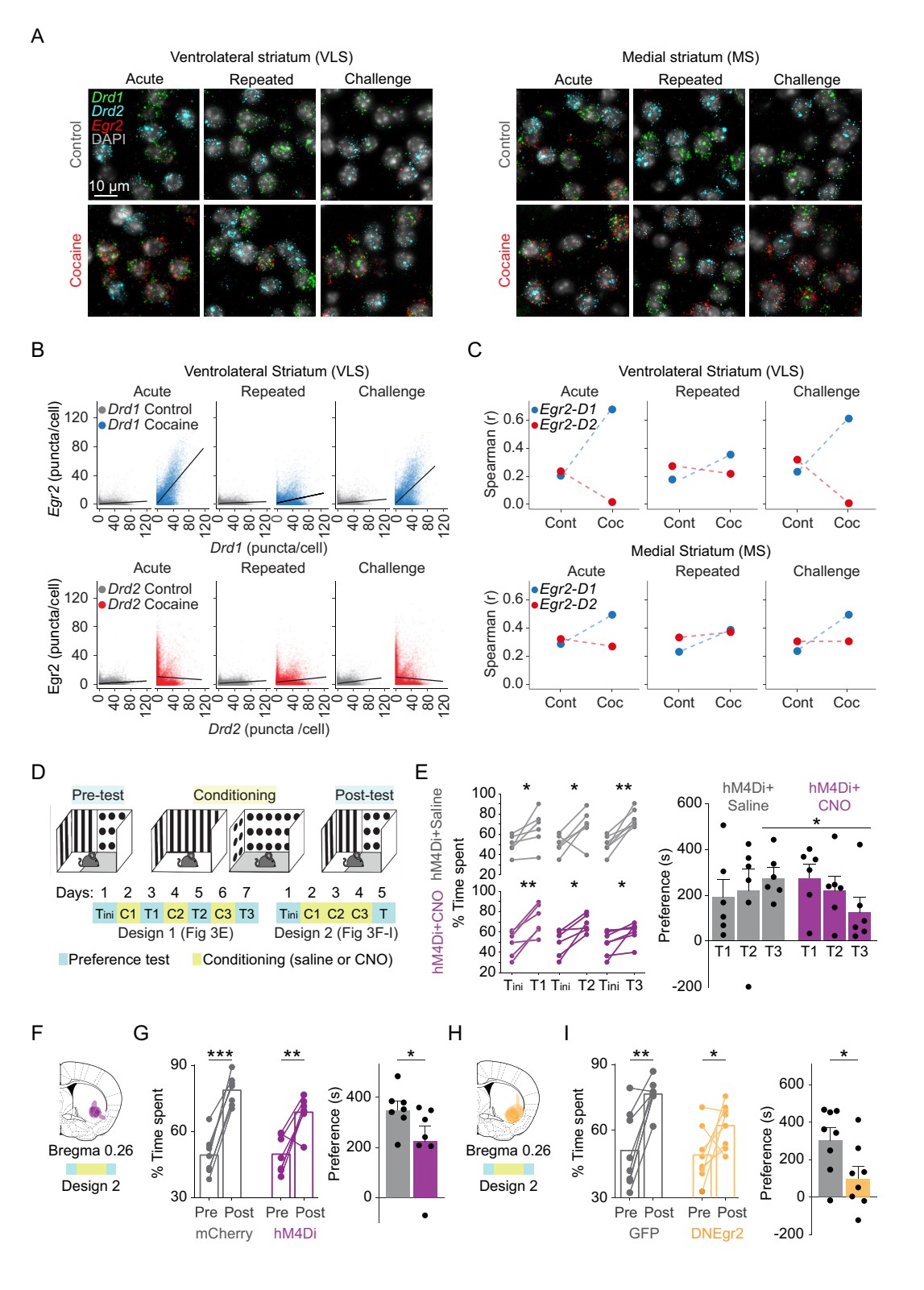

**Figure 3.** Induction of *Egr2* in VLS neurons contributes to the acquisition of cocaine reward. (**A**) Representative 40× images showing *Egr2* expression within *Drd1*[+] and *Drd2*[+] SPNs in the VLS (left) and MS (right) following acute, repeated, and challenge cocaine exposures compared to controls. (**B, C**) The *Drd1*[+] enriched IEG response in the VLS is dampened following repeated exposure to cocaine. (**B**) Scatter plots show cellular *Egr2* expression with *Drd1* or *Drd2* expression (puncta/cell) within individual cells. n = 6 sections from three mice for each condition (gray – 0 hr for either *Drd1* or *Drd2*

*Figure 3 continued on next page*

*Figure 3 continued*

combination, and blue or red – for *Drd1* or *Drd2* combination, respectively, 1 hr following cocaine experience). (*Drd1-Egr2*: acute control, slope = 0.028, $r^2$ = 0.038; acute cocaine, slope = 0.65, $r^2$ = 0.50; repeated control, slope = 0.019, $r^2$ = 0.02; repeated cocaine, slope = 0.11, $r^2$ = 0.1; challenge control, slope = 0.048, $r^2$ = 0.05; challenge cocaine, slope = 0.51, $r^2$ = 0.4. *Drd2-Egr2*: acute control, slope = 0.029, $r^2$ = 0.040; acute cocaine, slope = −0.036, $r^2$ = 0.0017; repeated control, slope = 0.03, $r^2$ = 0.069; repeated cocaine, slope = 0.06, $r^2$ = 0.03; challenge control, slope = 0.07, $r^2$ = 0.1; challenge cocaine, slope = −0.048, $r^2$ = 0.004. Pearson correlation, p<0.0001 for all conditions; refer to *Supplementary file 6* for detailed statistics. (**C**) Spearman correlation plots showing acute induction of *Egr2* is correlated with *Drd1* expression in the VLS, dampened following repeated exposure and re-emerges following challenge exposure. In the MS, *Egr2* expression is consistently correlated to both *Drd1* and *Drd2* expression following acute, repeated, and challenge exposures. Refer to *Figure 3—figure supplement 1* for additional correlations and *Figure 3—figure supplement 2* for reference to *Drd1* and *Drd2* expression levels throughout the study. (**D**) Scheme of experimental paradigms for testing conditioned-place preference (CPP) for cocaine. Mice were tested (cyan) for initial preference ('Tini') followed by either three interleaved pairs of conditioning (yellow) – test days ('Design 1', relevant for panel E) – or three consecutive conditioning days and then a final preference test ('Design 2', relevant for **F–I**). (**E**) Chemogenetic inhibition of VLS-Egr2 expressing neurons impairs cocaine CPP. Egr2-CRE animals were transduced with AAV-DIO-hM4Di-mCherry and, following 3 weeks of recovery, subjected to a paradigm of cocaine CPP in which the preference of mice was tested repeatedly following individual training days ('Design 1'; conditioning – days 2, 4, 6; tests – days 1, 3, 4, 7). The control group of mice was exposed to saline while experimental mice received CNO (5 mg/kg) 30 min prior to cocaine conditioning. Left – Line graphs representing % time spent on the cocaine paired side in individual preference test session (T_1, T_2, T_3) compared to the initial preference (initial preference test; T_ini). n = 6 mice in each group. *p<0.05, **p<0.01, ***p<0.005; paired t-test. Right – Bar graphs displaying the mean preference score (time spent on the drug paired side for relevant test session – initial test day). Significant difference in preference score is observed after three rounds of conditioning with cocaine n = 6 mice in each group. *p<0.05, **p<0.01, ***p<0.005; paired t-test. Data represented as mean ± sem. (**F**) Summary of expression domains of AAV-DIO-h4MDi in Egr2-CRE mice. (**G**) Chemogenetic inhibition of VLS-Egr2 expressing neurons during conditioning attenuates the development of cocaine CPP. Egr2-CRE animals were stereotactically transduced with AAV-DIO-mCherry (VLS-Egr2^mCherry) or AAV-DIO-hM4Di-mCherry (VLS-Egr2^hM4Di), and following recovery, all mice were subjected to cocaine CPP conditioning 30 min following exposure to CNO (10 mg/kg). Left panel represents change in % time spent on the cocaine paired side before and after conditioning for individual animals and the mean (paired t-test), while right panel (bar graphs) displays the mean preference score (time spent on the drug paired side of the final – first test day; unpaired t-test). Both groups developed CPP (paired t-test), while VLS-Egr2^hM4Di mice displayed a lower preference score compared to VLS-Egr2^mCherry controls (unpaired t-test). n = 7 mice in each group. *p<0.05, **p<0.01, ***p<0.005. For further documentation of expression domains and locomotion, see *Figure 3—figure supplement 3*. (**H**) Summary of expression domains of AAV-DN-Egr2. (**I**) Disruption of Egr2 function in the VLS inhibits the development of cocaine place preference. Left panel represents change in % time spent on the cocaine paired side before and after conditioning for individual animals and the mean, while bar graphs (right panel) display the mean preference score. Both groups developed CPP (paired t-test), while mice expressing AAV-DNEgr2 displayed a lower preference score compared to AAV-GFP controls (unpaired t-test). n = 8 mice in each group. *p<0.05, **p<0.01, ***p<0.005. For further documentation of expression domains, locomotion, and gene expression, see *Figure 3—figure supplement 4*. Images relating to acute cocaine (in **A**) were replicated from *Gonzales et al., 2020*, with permission.

The online version of this article includes the following figure supplement(s) for figure 3:

**Figure supplement 1.** Induced IEGs are correlated to *Drd1* expression in the VLS and to both *Drd1* and *Drd2* expressions in the MS.

**Figure supplement 2.** *Drd1* and *Drd2* receptor expression in the VLS and MS.

**Figure supplement 3.** DREADD inhibition of VLS Egr2^+ cells does not affect locomotion.

**Figure supplement 4.** Disruption of Egr2 function in the VLS does not affect locomotor behavior or IEG expression outside of VLS.

In a subsequent experiment, we implemented conditioning to the cocaine-associated context for three consecutive days prior to performing a preference test and exposed both experimental (hM4Di-expressing) and control (mCherry-expressing) groups to CNO (10 mg/kg; i.p.) prior to cocaine conditioning session (*Figure 3D*, 'Design 2'). We found that both groups demonstrated CPP (*Figure 3G*; paired t-test on % time spent on drug paired side; p_VLS-Egr2mCherry < 0.00001, t = −11.362, df = 6; p_VLS-Egr2hM4Di < 0.003, t = −4.1232, df = 6). However, mice in which VLS *Egr2*^+ neurons were inhibited (VLS-Egr2^hM4Di) displayed lower preference for the drug paired context (preference score) compared to control mice (VLS-Egr2^mCherry) (*Figure 3G*; p<0.05, t = 1.95, df = 9.4, one-tailed t-test). Importantly, no differences in locomotion were observed between groups on conditioning or test days (p=0.93, $F_{4,48}$ = 0.19, ANOVA; *Figure 3—figure supplement 3C*). We therefore conclude that VLS *Egr2*^+-expressing neurons contribute to the development of cocaine-seeking behavior, with no obvious impact on locomotor aspects of cocaine-driven behavior.

Salient experiences in general, and specifically exposure to cocaine, are thought to modify future behavior through induced gene expression responses, leading to stable changes in cell and circuit function (*Nestler and Lüscher, 2019*; *Robison and Nestler, 2011*). We hypothesized that the induction of *Egr2* by cocaine within VLS neurons may play a causal role in cocaine-induced modification of behavior. To assess a potential link between the expression of *Egr2* and cellular plasticity responsible for such behavioral modification, we ran an additional CPP experiment, following bilateral viral

transduction of the VLS neurons with AAV-eGFP (VLS$^{GFP}$), or a dominant-negative (S382R, D383Y) isoform of *Egr2* (VLS$^{DNEgr2}$; *Figure 3H*, *Figure 3—figure supplement 4A,B*). The dominant-negative mutation disrupts the DNA-binding activity of Egr2, while not interfering with the capacity of the protein to form heteromeric complexes with its natural binding partners, effectively inhibiting transcriptional activation of downstream genes regulated by Egr2 (*LeBlanc et al., 2007*; *Nagarajan et al., 2001*). Comparing the development of cocaine CPP, we found that both groups of mice developed CPP (*Figure 3I*, paired t-test on % time spent on drug paired side; $p_{VLS-GFP}$ <0.001, t = −4.9782, df = 7; $p_{VLS-DNEgr2}$ <0.05, t = −2.2199, df = 7). However, VLS$^{DNEgr2}$ developed lower CPP than VLS$^{GFP}$ mice (*Figure 3I*, p<0.05, t = 2.36, df = 14, unpaired t-test). No differences in locomotion were observed between the groups of mice (p=0.7, $F_{4,56}$ = 0.54, ANOVA; *Figure 3—figure supplement 4C*). These results assign a functional role to *Egr2* induction, primarily within VLS *Drd1*$^+$ neurons, in the development of conditioned-place preference to cocaine. To test the effect of disrupting Egr2 complexes may have on transcription, we analyzed the expression of *Arc*, *Egr2*, and *Nr4a1* in the VLS, MS, NAc, and LCtx. In the VLS, we observed the anticipated overexpression of *Egr2* (*Figure 3—figure supplement 4D*, $p_{(Egr2)}$<0.05, t = −5.3616, df = 2; two-tailed t-test, reflecting exogenous expression of the mutant gene), as well as blunted *Arc* and *Nr4a1* expression (*Figure 3—figure supplement 4D*, $p_{(Arc)}$<0.01, t = 6, df = 2.8; $p_{(Nr4a1)}$<0.005, t = 6.2, df = 3.8; two-tailed t-test). We did not observe any clear differences in gene expression between groups within other structures, demonstrating the localized effect of our viral manipulation (*Figure 3—figure supplement 4E–G*). These results demonstrate a role for cocaine-induced expression of *Egr2* in the VLS in supporting the development of cocaine-seeking and suggest that inducible transcriptional complexes involving Egr2 are functional in facilitating drug-induced maladaptive plasticity.

## Discussion

Drugs of abuse such as cocaine are known to act on key brain circuits, modifying and biasing the future behavior of an individual toward increased drug seeking. In this study, we develop a comprehensive compendium of the transcriptional dynamics induced within key brain regions during the development of cocaine sensitization. We highlight the striatum as a major hub of plasticity, within which we identify differential transcriptional recruitment of neuronal ensembles by cocaine, dependent on striatal subdomain, identity of projection neurons and the history of cocaine exposure. Finally, we focus on a prominent cocaine-sensitive IEG, *Egr2*, and show that *Egr2*-expressing SPNs in the VLS, and the expression of *Egr2* within them, support drug-seeking behavior.

Repeated exposure to cocaine, as well as abstinence, produces long-lasting functional changes in the reward circuit to drive the maladaptive modification of reinforced behavior (*Dong and Nestler, 2014*; *Everitt, 2014*; *Gremel and Lovinger, 2017*; *Hyman et al., 2006*; *Kelley, 2004*; *Lüscher, 2016*; *Lüscher and Malenka, 2011*; *Nestler, 2013*; *Russo and Nestler, 2013*; *Salery et al., 2020*; *Volkow and Morales, 2015*; *Wolf, 2016*; *Zahm et al., 2010*). The imprinting of such potentially lifelong alterations in behavior driven by drug experience is supported by cocaine-induced modifications in gene expression (*McClung and Nestler, 2008*; *Nestler, 2002*; *Nestler and Lüscher, 2019*; *Salery et al., 2020*; *Steiner, 2016*; *Steiner and Van Waes, 2013*). In this study, using an unbiased approach to screen gene expression, we resolved the transcriptional landscapes of distinct cocaine experiences across multiple reward-related brain circuits with broad temporal resolution. Our approach allowed us to describe transcripts modulated at updated baselines (after a history of either repeated cocaine exposure or abstinence), as well as in the hours following exposure to distinct cocaine experiences.

Baseline transcriptional changes in cortical and basal ganglia structures following defined cocaine schedules have been described previously in both rodents and humans (*Bannon et al., 2005*; *Bannon et al., 2014*; *Eipper-Mains et al., 2013*; *Freeman et al., 2010*; *Gao et al., 2017a*; *Hurd and Herkenham, 1993*; *Lull et al., 2008*; *Ribeiro et al., 2017*; *Walker et al., 2018*). Consistent with previous findings, we observed dynamic shifts in baseline gene expression in multiple categories potentially associated with neuronal plasticity (synaptic genes; genes associated with projection morphogenesis, actin filament regulation; proteostasis and myelin). Interestingly, genes associated with neuronal morphology and synaptic function demonstrated unique patterns of shifts within different brain structures. For example, expression of genes such as *Vamp*, *Pkrcg*, *Ncdn*,

*Camk2b, Shank3*, and *Syp* were downregulated in the NAc following repeated cocaine exposure, while being upregulated in the DS. Such region-specific shifts in gene expression may support circuit-specific structural and functional modifications to cell assemblies (*Clayton et al., 2020*; *Kyrke-Smith and Williams, 2018*). Myelin genes (*Plp1, Mobp, Mbp, Mal, Pllp*) were downregulated across all structures studied (LCtx, Amy, NAc, DS, LH), initially following repeated cocaine exposure, and further following abstinence, across all experimental mice. Conversely, genes associated with proteostasis (e.g., chaperones such as members of the CCT, Hsp40, Hsp70, and Hsp90 complexes) were upregulated in concert across structures following cocaine abstinence. Notably, similar changes in myelin genes and genes associated with proteostasis have been described in both human and rodent studies (*Albertson et al., 2004*; *Bannon et al., 2014*; *García-Fuster et al., 2012*; *Johnson et al., 2012*; *Kovalevich et al., 2012*; *Lull et al., 2008*; *Narayana et al., 2014*), but their functional implications remain unknown. Future investigation into the features of cocaine experience-related transcriptome is anticipated to provide targets for intervention, potentially supporting the reversal of brain function to a 'cocaine-naive' state.

IEG expression is well accepted to be the substrate for long-term modulations supporting memory formation (*Alberini, 2009*; *Alberini and Kandel, 2015*). Although cocaine-induced IEG expression has been extensively characterized in rodents (*Berke et al., 1998*; *Caster and Kuhn, 2009*; *Gao et al., 2017a*; *Guez-Barber et al., 2011*; *Moratalla et al., 1996*; *Piechota et al., 2010*; *Robison and Nestler, 2011*; *Savell et al., 2020*; *Steiner, 2016*; *Valjent et al., 2006*; *Zahm et al., 2010*), these studies were mostly limited in the number of genes analyzed and restricted to isolated brain structures following specific drug regimens. Addressing the cocaine-induced transcriptome, we observed transcriptional recruitment of the LCtx, Amy, NAc, and DS, of which the DS was most prominent. Furthermore, the immediate-early transcriptional programs induced across other tissues largely consisted of subcomponents of DS programs. What does this imply? We propose thathe overlapping fraction of induced genes is representative of a 'core transcriptome' that is consistently induced across many structures or cell types and only varies in the magnitude of their expression (*Hrvatin et al., 2018*; *Savell et al., 2020*; *Tyssowski et al., 2018*). This core component predominantly corresponds to signaling molecules and transcriptional regulators (the genes common across most programs are *Arc, Arl4d, Btg2, Ddit4, Dusp1, Egr2, Egr4, Fos, Fosb, Junb, Nr4a1, Per1*, and *Tiparp*), likely responsible for transforming inducing signals into instructions for implementation of appropriate synaptic, cellular, and circuit-specific plasticity mechanisms by 'effector' genes. These downstream effector genes are induced in a secondary wave of transcription, corresponding to the significantly diversified gene response at 2–4 hr following cocaine (*Amit et al., 2007*; *Clayton et al., 2020*; *Gray and Spiegel, 2019*; *Hrvatin et al., 2018*; *Mukherjee et al., 2018*; *Tyssowski et al., 2018*; *Yap and Greenberg, 2018*). Interestingly, a recent landmark study (*Savell et al., 2020*) utilized a multiplexed CRISPR strategy to drive co-expression of genes overlapping with many of the components of the putative 'core transcriptome' (*Btg2, Egr2, Egr4, Fos, FosB, JunB*, and *Nr4a1*) in the NAc and found that this manipulation increased SPN excitability and enhanced the development of cocaine sensitization.

What might be the role of the transcriptional induction in the DS and its subsequent dampening? It is becoming more broadly accepted that IEG induction serves to support long-term plasticity (*Chandra and Lobo, 2017*; *Clayton, 2000*; *Clayton et al., 2020*; *Mukherjee et al., 2018*; *Tyssowski and Gray, 2019*). The MS is defined as the 'associative striatum' and is associated with goal-directed behaviors, as well as defining the vigor of locomotor actions (*Balleine and O'Doherty, 2010*; *Balleine and Ostlund, 2007*; *Kravitz et al., 2010*; *Lipton et al., 2019*; *Nonomura et al., 2018*). We propose that the cocaine-driven locomotor sensitization may be mediated by the balanced and largely maintained transcriptional induction within Drd1/Drd2 SPNs in the MS. The lateral 'sensori-motor' striatum is strongly associated with habit formation and compulsive drug seeking (*Lipton et al., 2019*; *Yin et al., 2004*; *Zapata et al., 2010*). Moreover, the VLS receives selective sensorimotor afferents mapped to upper limb and orofacial cortical regions. Interestingly, behavioral stereotypies, primarily upper limb and orofacial, arise upon high-dose psychostimulant exposure (*Karler et al., 1994*; *Murray et al., 2015*; *Schlussman et al., 2003*), and orofacial stereotypies have been induced following selective infusion of psychostimulants to the VLS (*Baker et al., 1998*; *Delfs and Kelley, 1990*; *Rebec et al., 1997*; *White et al., 1998*). It is intriguing to consider the possibility that recruitment of plasticity mechanisms within VLS Drd1$^+$ neurons supports the increased propensity to engage in orofacial stereotypies, while the subsequent dampening of cocaine-induced

transcription within these neurons may indicate the 'canalization' of this limited action repertoire, at the expense of a broader behavioral repertoire. This topic will form the basis for future investigation.

Infusion of psychostimulants into the VLS has also been shown to promote operant reinforcement and conditioned-place preference, implicating it in reward and reinforcement (*Baker et al., 1998*; *Kelley and Delfs, 1991*). In order to query the role of the VLS IEG-expressing ensembles in the development of cocaine context association, we inhibited the activity of VLS *Egr2*⁺ neurons by conditional expression of hM4Di DREADDs, which curbed CPP. To directly investigate a role for VLS IEG induction on CPP behavior, we expressed a dominant-negative isoform of Egr2 (in which the DNA-binding domain was inactivated) in the VLS and observed a similar impact. Thus, to our knowledge, we provide the first functional implication of the VLS in cocaine seeking. Furthermore, we describe cellular dynamics of transcriptional recruitment of VLS IEG⁺ neurons (primarily *Drd1*⁺) during the development of behavioral sensitization to cocaine. The development and execution of drug-seeking behavior is heavily context dependent (*Calipari et al., 2016*; *Crombag et al., 2002*; *Crombag et al., 2008*; *Crombag and Shaham, 2002*; *Cruz et al., 2014*; *Lee et al., 2006*; *Rubio et al., 2015*). Potentially, the dampening of sensorimotor VLS IEG induction following repeated cocaine could serve to 'cement' the initial context association, limiting behavioral flexibility and the capacity to revert context association, exacerbating the impact of contextual cues on drug seeking behavior (*Calipari et al., 2016*; *Crombag and Shaham, 2002*; *Gipson et al., 2013*; *Hyman, 2005*; *Phillips et al., 2003*; *Shaham et al., 2003*; *Volkow et al., 2006*).

Recently, we have shown that salient experiences are represented in the mouse brain by unique patterns of gene expression. Thus, the induction pattern of a handful of genes is sufficient to decode the recent experience of individual mice with almost absolute certainty. Of these, the IEG whose expression contributes most towards classification of the recent experience of individual mice is *Egr2* (*Mukherjee et al., 2018*). *Egr2* is, furthermore, the gene most robustly induced by cocaine in the dorsal striatum (*Gonzales et al., 2020*; *Mukherjee et al., 2018*; *Supplementary file 4*) and is a sensitive indicator of cocaine-engaged striatal cell assemblies (*Gonzales et al., 2020*). In the current study, we initiated investigation into the role of *Egr2* in promoting drug seeking. Previous studies have shown that Egr2 is crucial for normal hindbrain development, peripheral myelination, and humoral immune response and is implicated in diseases such as congenital hypomyelinating neuropathy, Charcot–Marie-Tooth disease, Dejerine–Sottas syndrome, as well as schizophrenia (*Boerkoel et al., 2001*; *De and Turman, 2005*; *Li et al., 2019*; *Morita et al., 2016*; *Okamura et al., 2015*; *Svaren and Meijer, 2008*; *Topilko et al., 1994*; *Warner et al., 1999*; *Warner et al., 1998*; *Wilkinson, 1995*; *Yamada et al., 2007*). In the central nervous system, *Egr2* has been shown to be induced by seizure activity, kainic acid injection, LTP-inducing stimuli in hippocampal neurons, as well as following administration of several groups of drugs such as methamphetamine, cocaine, heroin, and alcohol (*Gao et al., 2017a*; *Gass et al., 1994*; *Imperio et al., 2018*; *López-López et al., 2017*; *Mataga et al., 2001*; *Rakhade et al., 2007*; *Saint-Preux et al., 2013*; *Worley et al., 1993*). However, the role *Egr2* may play in encoding memory or drug-induced behavior remained unresolved. Our findings show that the activity of *Egr2* is required for the full development of cocaine place preference, and highlight an additional member of the Egr family, alongside *Egr1* and *Egr3*, in drug-induced plasticity (*Bannon et al., 2014*; *Chandra et al., 2015*; *Moratalla et al., 1992*; *Valjent et al., 2006*). *Egr2* has been implicated in the regulation of cell-specific gene expression in peripheral Schwann cells (*Jang et al., 2006*) and fibroblasts (*Fang et al., 2011*), and disruptions to Egr2 DNA binding have been implicated in diseases of myelination and brain development. However, we are not aware of any study identifying the targets of Egr2 in the mature brain. We report downregulated expression of *Nr4a1* and *Arc* following overexpression of dominant-negative Egr2 in the VLS. However, as we did not identify Egr2 binding sites within regulatory regions of *Nr4a1* or *Arc*, we hypothesize that the impact of DN-Egr2 expression on *Nr4a1* and *Arc* may be indirect, a point for future investigation.

In conclusion, our study provides (1) a comprehensive description of brain-wide transcriptional dynamics, as well as spatial dynamics of SPN-specific IEG recruitment during the development of cocaine sensitization and (2) a demonstration of the role of VLS *Egr2*-expressing ensembles, as well as VLS expression of *Egr2*, in the development of cocaine seeking. Future work will address the mechanisms supporting cell-type specificity of transcriptional induction, as well as the role of IEG-mediated plasticity mechanisms in VLS-dependent stereotypy and context association.

# Materials and methods

## Key resources table

| Reagent type (species) or resource | Designation | Source or reference | Identifiers | Additional information |
|---|---|---|---|---|
| Strain, strain background (*Mus musculus*) | Wild-type C57BL/6OLAHSD mice | The Harlan Laboratory | NA | |
| Strain, strain background (*Mus musculus*) | Egr2-Cre knock in mice | The Jackson Laboratory | Cat# 025744 RRID: IMSR_JAX:025744 | |
| Recombinant DNA reagent | AAV2-hSyn-DIO-hM4d(Gi)-mCherry | Addgene | Cat# 44362-AAV2 RRID: Addgene_44362 | 1.15 dilution |
| Recombinant DNA reagent | AAV2-hSyn-DIO-mCherry | UNC vector core facility | N/A | 1.15 dilution |
| Recombinant DNA reagent | AAVdj-CMV-eGFP | ELSC vector core facility | N/A | 1.15 dilution |
| Recombinant DNA reagent | AAVdj-CAG-DNEgr2-IRES-GFP | ELSC vector core facility | N/A | 1.15 dilution |
| Recombinant DNA reagent | Plasmid with dominant negative mutant Egr2 (S382R,D383Y) | Jeffrey Milbrant, Washington University | N/A | |
| Chemical compound, drug | Clozapine-N-oxide (CNO) | Sigma–Aldrich | Cat # C0832-5MG | |
| Chemical compound, drug | Cocaine | Hadassah Hospital Pharmacy | N/A | |
| Commercial assay, kit | Fluorescent Multiplex Reagent Kit | Advanced Cell Diagnostics RNAscope | Cat # 320850 | |
| Commercial assay, kit | NEBNext Ultra II Non-Directional RNA Second-Strand Synthesis Module | New England Biolabs | Cat # E6111L | |
| Commercial assay, kit | KAPA Hifi Hotstart ReadyMix | Roche | Cat # KK-KK2601-2 07958927001 | |
| Commercial assay, kit | MinElute Gel Extraction Kit | Qiagen | Cat # 28604 | |
| Commercial assay, kit | NEBNext Library Quant Kit for Illumina | New England Biolabs | Cat # E7630L | |
| Commercial assay, kit | High-sensitivity DNA kit | Agilent Technologies | Cat # 5067–4626 | |
| Commercial assay, kit | NextSeq 500 High Output V2 kits | Illumina | Cat # FC-404–2005 | |
| Commercial assay, kit | SMARTScribe Reverse Transcriptase | Takara | Cat # 639536 | |
| Sequence-based reagent (smFISH) | Probe-Mm-Drd1a-C2 | Advanced Cell Diagnostics RNAscope | Cat # 406491-C2 | |

*Continued on next page*

*Continued*

| Reagent type (species) or resource | Designation | Source or reference | Identifiers | Additional information |
|---|---|---|---|---|
| Sequence-based reagent (smFISH) | Probe-Mm-Drd1a-C3 | Advanced Cell Diagnostics RNAscope | Cat # 406491-C3 | |
| Sequence-based reagent (smFISH) | Probe-Mm-Drd2-C2 | Advanced Cell Diagnostics RNAscope | Cat # 406501-C2 | |
| Sequence-based reagent (smFISH) | Probe-Mm-Egr2 | Advanced Cell Diagnostics RNAscope | Cat # 407871 | |
| Sequence-based reagent (smFISH) | Probe-Mm-Fos-C3 | Advanced Cell Diagnostics RNAscope | Cat # 316921-C3 | |
| Sequence-based reagent (smFISH) | Probe-Mm-Arc-C3 | Advanced Cell Diagnostics RNAscope | Cat # 316911-C3 | |
| Sequence-based reagent (smFISH) | Probe-Mm-Nr4a1-C2 | Advanced Cell Diagnostics RNAscope | Cat # 423341-C2 | |
| Sequence-based reagent (RNA-seq) | Primers for first-strand synthesis | This paper | N/A | CGATTGAGGCCGGTAATACGACTCACTAT AGGGGCGACGTGTGCTCTTCCGATCT NNNNNNNNNNNNNNNTTTTTTT TTTTTTTTTTTTN |
| Sequence-based reagent (RNA-seq) | Forward primer with P5-Read1 sequence | This paper | NA | AATGATACGGCGACCACCGAGATCTACA CTAGATCGCTCGTCGGCAGCGTCAGATGTG |
| Sequence-based reagent (RNA-seq) | Reverse primer with P7-Read2 sequence | This paper | NA | CAAGCAGAAGACGGCATACGAGATGT GACTGGAGTTCAGACGTGTGCTCTTCCGATCT |
| Software, algorithm | R | R studio | https://rstudio.com/products/rstudio/ | |
| Software, algorithm | ImageJ | National Institutes of Health | https://imagej.nih.gov/ij/ RRID:SCR_003070 | |
| Software, algorithm | CellProfiler | Broad Institute | https://cellprofiler.org/ RRID:SCR_007358 | |
| Software, algorithm | Prism7 | GraphPad | https://www.graphpad.com RRID:SCR_002798 | |
| Software, algorithm | Ethovision XT | Noldus | https://www.noldus.com/ethovision-xt RRID:SCR_000441 | |
| Software, algorithm | Photoshop and Illustrator | Adobe | https://www.adobe.com/in/ creativecloud/catalog/desktop.html? promoid=PTYTQ77P&mv=other | |
| Other | 0.9% Nacl | | Cat # 3642828 | |
| Other | Isoflurane | Piramal Critical Care | Cat # AWN34014604 | |
| Other | Microtome (7000 smz2) | Camden Instruments | https://www.emsdiasum.com/ microscopy/products/equipment/ vibrating_microtome.aspx | |
| Other | Stereoscope | Olympus | Cat # N1197800 | |
| Other | TissueLyser LT | Qiagen | Cat # 69980 | |
| Other | Superfrost Plus slides | ThermoFisher Scientific | Cat # J1800AMNZ | |

*Continued on next page*

*Continued*

| Reagent type (species) or resource | Designation | Source or reference | Identifiers | Additional information |
|---|---|---|---|---|
| Other | Hermes high-definition cell-imaging system | Wiscan | https://idea-bio.com/products/wiscan-hermes/ | |
| Other | SomnoSuite Low-Flow Anesthesia System | Kent Scientific Corporation | https://www.kentscientific.com/products/somnosuite/ | |
| Other | Fine drill burr | RWD Life Science | Cat # 78001 | |
| Other | Microsyringe (33G) | Hamilton | Cat # 65460–05 | |
| Other | 3M Vetbond tissue Adhesive | 3M (Ebay) | Cat # 8017242664 | |
| Other | Isoflurane | Piramal Critical Care | Cat # AWN34014604 | |
| Other | Tri-Reagent | Sigma–Aldrich | Cat # T9424 | |
| Other | OCT embedding medium | Scigen Scientific Gardena | Cat # 23-730-625 | |
| Other | ACD RNAscope fresh frozen tissue pretreatment | Advanced Cell Diagnostics RNAscope | Cat # 320513 | |
| Other | DAPI | Sigma–Aldrich | Cat # 10236276001 | |
| Other | Lab Vision PermaFluor Aqueous Mounting Medium | ThermoFisher Scientific | Cat # TA-030-FM | |
| Other | dNTPs | New England Biolabs | Cat # N0447s | |
| Other | $MnCl_2$ | Sigma–Aldrich | Cat # 244589–10G | |
| Other | SPRI magnetic beads | Beckman Coulter | Cat # A63881 | |
| Other | 1 M Tris–HCl, pH 8.0 | ThermoFisher Scientific | Cat # 15568025 | |
| Other | SDS Solution (10%) | Biological Industries | Cat # 01-890-1B | |

## Lead contact and materials availability

Further information and requests for resources should be directed to and will be fulfilled by the Lead Contact, Ami Citri (ami.citri@mail.huji.ac.il). This study did not generate new unique reagents.

## Experimental models and subject details

Male C57BL/6OLAHSD mice used for RNA-sequencing and single-molecule FISH analysis following cocaine sensitization were obtained from Harlan Laboratories, Jerusalem, Israel. Transgenic Egr2-Cre knock-in mice were obtained from Jackson Laboratories. All animals were bred at Hebrew University, Givat Ram campus, by crossing positive males with C57BL/6OLAHSD female mice obtained from Harlan Laboratories. All animals (wild types and transgenic littermates of same sex) were group housed both before and during the experiments. They were maintained under standard environmental conditions – temperature (20–22℃), humidity (55 ± 10%), and 12–12 hr light/dark cycle (7 am on and 7 pm off), with ad libitum access to water and food. Behavioral assays were performed during the light phase of the circadian cycle. All animal protocols (# NS-13-13660-3; NS-13-13895-3; NS-15-14326-3; NS-16-14644-2; NS-14667–3; NS-16-14856-3; NS-19-15753-3) were approved by the Institutional Animal Care and Use Committees at the Hebrew University of Jerusalem and were in

accordance with the National Institutes of Health Guide for the Care and Use of Laboratory Animals. Animals were randomly assigned to individual experimental groups, with some exceptions, such as in case of conditioned-place preference experiments (elaborated later). Experimenters were blinded regarding experimental manipulations wherever possible. While all experiments were performed in male mice, we do not anticipate that the results would differ between males and females, as similar gene programs are recruited in both (*Savell et al., 2020*).

| Animals | Sex | Age (weeks) |
| --- | --- | --- |
| Wild-type C57BL/6 mice | Male | 6–7 |
| Egr2-Cre knock in mice | Male | 10–30 |

## Detailed methods
### Behavioral assays
#### Cocaine sensitization

Six to seven week old C57BL/6OLAHSD mice, after arriving from Harlan Laboratories, were first allowed to acclimate to the SPF facility for a period of 5–7 days. Animals were then briefly handled once or twice daily for 2–3 days. During the handling sessions, animals were allowed to freely move around on the experimenter's palm for 1–2 min either alone or in pairs. On the following three consecutive days, mice were subjected to once daily intraperitoneal (IP) saline injections (250 µl) and immediately transferred to a clear Plexiglas box (30 × 30 × 30 cm) within a sound- and light-attenuated chamber fitted with an overhead camera, for ~20 min, and then returned to their home cage. After this habituation phase, animals were subjected to one daily IP cocaine injection (20 mg/kg; Stock solution: 2 mg/ml dissolved in 0.9% saline and injected at 10 ml/kg volume), according to the following groups: (1) *acute cocaine* group received a single dose of IP cocaine, (2) *repeated cocaine* group was administered cocaine once daily for five consecutive days, and (3) *challenge cocaine* group of animals was treated similarly to the repeated cocaine group for the first 5 days, subjected to abstinence (no drug treatment) for 21 days, and then re-exposed to a single dose of cocaine. Animals sacrificed directly from the home cage without any treatment were regarded as controls in the experiment (0 hr) and interleaved with the other groups corresponding to the relevant cocaine regiment (acute, chronic, and challenge cocaine). Transcription was analyzed at 1, 2, and 4 hr following the cocaine injection for the RNA-seq experiments. In smFISH experiments, animals were sacrificed for brain collection 1 hr after the cocaine injection, while control animals were treated as described earlier. Locomotor activity was measured as distance traveled in the open field arena for a period of 15 min, following either saline/cocaine injections, on each day was quantified by Ethovision (Noldus) software.

#### Conditioned-place preference

Conditioned-place preference was assessed in a custom-fitted arena (Plexiglass box [30 × 30 × 30 cm]) designed in-house and placed in individual light- and sound-attenuated chambers as in *Terem et al., 2020*. On the preference test days, the arena was divided into two compartments of equal dimensions. One compartment was fitted with rough floor ('crushed ice' textured Plexiglas) and black (on white) dotted wallpaper, while the other was fitted with smooth floor with black (on white) striped wallpaper. On the conditioning days, animals were presented with only one context in each training session, such that the entire box had rough flooring and dotted wallpaper or smooth flooring with striped wallpaper. Animals were placed in the center of the arena, and free behavior was recorded for 20 min. General activity and position/location of the mice in the arena were monitored by video recording using an overhead camera. Baseline preference was measured using the Ethovision XT software by analyzing the time spent in each chamber during the 20 min session. Mice were randomly assigned a conditioning compartment in order to approximately balance any initial bias in preference toward a specific chamber. *Procedure:* All experiments were performed using an unbiased design and consisted of the following phases: Handling: Two to three days performed twice daily and involved free exploration on the palms of the experimenter for 2–3 min. Pre-test: Single 20 min session (performed around noon), during which animals explored the arena which was

divided into two compartments. Conditioning: Three days of two counterbalanced 20 min sessions per day separated by at least 4 hr. Mice were randomly assigned to a context (combination of a single floor-type and wallpaper patterns, as described above), which was paired with IP injections of saline (250 µl), and a separate context, which was paired with IP cocaine (10 mg/kg; Stock solution: 1 mg/ml dissolved in 0.9% saline and injected at 10 ml/kg volume). Post-conditioning final preference test was performed as in the pre-test.

For chemogenetic experiments, CNO (10 mg dissolved in 500 µl DMSO and then mixed into 9.5 ml 0.9% saline, to a total of 10 ml CNO solution at a concentration of 1 mg/ml) was injected at a dose of 5 or 10 mg/kg 30 min before cocaine conditioning sessions.

## Tissue dissections and RNA extraction

Collection of tissue samples (*Figure 3—figure supplement 1*) and RNA extraction were performed as described previously (*Mukherjee et al., 2018*; *Turm et al., 2014*), with few modifications. Briefly animals were anesthetized in isoflurane (Piramal Critical Care), euthanized by cervical dislocation, and the brains quickly transferred to ice-cold artificial cerebrospinal fluid (ACSF) solution. Coronal slices of 400 µm were subsequently made on a vibrating microtome (7000 smz2; Camden Instruments) and relevant brain areas dissected under a stereoscope (Olympus). Tissue pieces were collected in PBS, snap-frozen in dry-ice, and on the same day transferred to Tri-Reagent (Sigma–Aldrich). The tissue was stored at −80°C until being processed for RNA extraction. For RNA extraction, the stored tissue was thawed at 37°C using a drybath and then immediately homogenized using TissueLyser LT (Qiagen). RNA extraction was performed according to the manufacturer's guidelines. All steps were performed in cold conditions.

## RNA-seq library preparation

One hundred nanogram of RNA was used for first-strand cDNA preparation as follows: The RNA was mixed with RT primers containing barcodes (seven bps) and unique molecular identifiers (UMIs; eight bps) for subsequent de-multiplexing and correction for amplification biases, respectively. The mixture was denatured in a Thermocycler (Bio-Rad) at 72°C for 3 min and transferred immediately to ice. An RT reaction cocktail containing 5× SmartScribe buffer, SmartScribe reverse transcriptase (Takara), 25 mM dNTP mix (NEB), and 100 mM MnCl$_2$ (Sigma) was added to the RNA and primer mix and incubated at 42°C for 1 hr followed by 70°C for 15 min. The cDNA from all samples were pooled, cleaned with 1.2× AMPURE magnetic beads (Beckman Coulter), and eluted with 10 mM Tris of pH 8 (ThermoFisher Scientific). The eluted cDNA was further processed for double-stranded DNA synthesis with the NEBNext Ultra II Non-Directional RNA Second-Strand Synthesis Module (NEB), followed by another round of clean-up with 1.4× SPRI magnetic beads. The resultant double-stranded cDNA was then incubated with Tn5 tagmentase enzyme and a 21 bp oligo (TCG TCGGCAGCGTCAGATGTG sequence) at 55°C for 8 min. The reaction was stopped by denaturing the enzyme with 0.2% SDS (Biological Industries), followed by another round of cleaning with 2× SPRI magnetic beads. The elute was amplified using the KAPA Hifi Hotstart ReadyMix (Kapa Biosystems along with forward primer that contains Illumina P5-Read1 sequence) and reverse primer containing the P7-Read2 sequence. The resultant libraries were loaded on 4% agarose gel (Invitrogen) for size selection (250–700 bp) and cleaned with Mini Elute Gel Extraction kit (Qiagen). Library concentration and molecular size were determined with NEBNext Library Quant Kit for Illumina (NEB) according to manufacturer's guidelines, as well as Bioanalyzer using High-Sensitivity DNA kit (Agilent Technologies). The libraries were run on the Illumina platform using NextSeq 500 High Output V2 kits (Illumina).

## Single-molecule fluorescence in-situ hybridization

A detailed protocol is available in *Gonzales et al., 2020*. Briefly, smFISH protocol was performed on 14 µm tissue sections using the RNAscope Multiplex Fluorescent Reagent kit (Advanced Cell Diagnostics) according to the RNAscope Sample Preparation and Pretreatment Guide for Fresh Frozen Tissue and the RNAscope Fluorescent Multiplex Kit User Manual (Advanced Cell Diagnostics). Image acquisition was performed using a Hermes high-definition cell-imaging system with 10 × 0.4 NA and 40 × 0.75 NA objectives. Five Z-stack images were captured for each of four channels – 475/28 nm (FITC), 549/15 nm (TRITC), 648/20 nm (Cy5), and 390/18 nm (DAPI). Image processing was performed using ImageJ software. Maximum-intensity images for each channel were obtained using

Maximum Intensity Z-projection. All channels were subsequently merged, and the dorsal striatum region was manually cropped from these merged images according to the Franklin and Paxinos Mouse brain atlas, Third edition. Quantification of RNA expression from images was done using the CellProfiler (*McQuin et al., 2018*) speckle counting pipeline.

## Stereotactic surgeries

Induction and maintenance of anesthesia during surgery were achieved using SomnoSuite Low-Flow Anesthesia System (Kent Scientific Corporation). Following induction of anesthesia, animals were quickly secured to the stereotaxic apparatus (David KOPF instruments). The skin was cleaned with Betadine (Dr. Fischer Medical), and Lidocaine (Rafa Laboratories) was applied to minimize pain. An incision was made to expose the skull, which was immediately cleaned with hydrogen peroxide (GADOT), and a small hole was drilled using a fine drill burr (RWD Life Science). Using a microsyringe (33G; Hamilton) connected to an UltraMicroPump (World Precision Instruments), virus was subsequently injected at a flow rate of 100 nl/min. Upon completion of virus delivery, the microsyringe was left in the tissue for up to 5 min and then slowly withdrawn. The skin incision was closed using a Vetbond bioadhesive (3M), the animals were removed from the stereotaxic apparatus, injected with saline and pain-killer Rimadyl (Norbrook), and allowed to recover under gentle heating. Coordinates of the stereotactic injection were determined using the Paxinos and Franklin mouse brain atlas. Every virus used in the study was titrated appropriately to ensure localized infections. All injections were performed bilaterally and observed to be symmetric.

## Coordinates of the stereotactic injection

| Experiment ID | Viruses | Coordinates | Strain | Virus expression time (days) |
|---|---|---|---|---|
| Chemogenetic inhibition (*Figure 3E*) | AAV2-hSyn-DIO-hM4d(Gi)-mCherry (n = 6; received saline) AAV2-hSyn-DIO-hM4d(Gi)-mCherry (n = 6, received CNO at 5 mg/kg) | AP: 0.9; ML: ±2.6; DV: 3.6 | Egr2-Cre | 21 |
| Chemogenetic inhibition (*Figure 3F–G*, *Figure 3—figure supplement 3*) | AAV2-hSyn-DIO-hM4d(Gi)-mCherry (n = 8; all received CNO at 10 mg/kg) AAV2-hSyn-DIO- mCherry (n = 8, all received CNO at 10 mg/kg) | AP: 0.9; ML: ±2.6; DV: 3.6 | Egr2-Cre | 21 |
| DN-Egr2 (*Figure 3H,I*, *Figure 3—figure supplement 4*) | AAVdj-CMV-eGFP (n = 8) AAVdj-CAG-DNEgr2-IRES-GFP (n = 8) | AP: 0.9; ML: ±2.65; DV: 3.6 | WT | 21 |

## Quantification and statistical analysis

### Statistical analysis and data visualization

R version 3.4.4 was used for all statistical analysis and graphical representations. Venn diagrams were generated with 'eulerr' package. Three-dimensional plots were generated with 'plot3D' package. Heatmaps were generated with 'Heatmap.2' function form 'gplots' package. All other figures were generated using 'ggplot2'. Details of the statistics applied in analysis of smFISH and behavioral experiments are summarized in *Supplementary file 6*.

## RNA-seq analysis

### Alignment and QC

RNA-seq read quality was evaluated using FastQC. PCR duplicates were removed using unique molecular identifiers (UMIs), and polyA tail, if existing, was trimmed from the 3' end of the reads. Reads were aligned to the mouse genome (GRCm38) using STAR, and HTseq was used to count the number of reads for each gene. Samples with less than 1 million usable reads were removed from

the analysis. Samples with more than 8 million reads were down-sampled to 50% (using R package 'subSeq'). The list of the samples analyzed in this paper and the distribution of library size are presented in *Supplementary file 1* and *Figure 1—figure supplement 2*. All raw sequencing data is available on NCBI GEO: GSE158588.

## Analysis of shifts in baseline transcription
In order to compare baseline shifts in gene expression following repeated cocaine administration, we compared gene expression within the samples obtained at time 0 (not exposed to cocaine on day of sample collection) in each one of the conditions – acute, repeated, and challenge cocaine (*Figure 1—figure supplement 3A* – heatmap of all genes exhibiting change). This analysis was performed with 'DEseq2' package in R. We used the Wald test in the DEseq function and compared gene expression in cocaine naïve mice vs. mice exposed to repeated cocaine, as well as comparing to abstinent mice following repeated cocaine. List of detected genes, normalized counts, and p-values (FDR corrected) are presented in *Supplementary file 2*. We observed that in a few samples, an apparent sequencing batch effect was detected, likely related to the library preparation and/or to the association of samples with different sequencing runs. Therefore, we performed the final analysis on only a subset of the samples, which did not exhibit a batch effect. While gene selection was performed on the subset of samples, the data portrayed in *Figure 1C* depicts all samples from the relevant time points – demonstrating that the genes identified from the subset of samples are consistently modified across all samples. Therefore, our gene list likely provides a conservative estimate of the true magnitude of shifts in gene expression.

## Analysis of inducible transcription
Detection of the induced genes following cocaine administration was performed with the 'DEseq2' package in R. Each structure was analyzed separately. The model included time (0, 1, 2, 4 hr after cocaine administration) and the experiment (acute, repeated, and challenge), as well as the interaction time × experiment. We used a likelihood ratio test (LRT) and selected genes changing over time in at least one of the experiments (eliminating genes that are changing only between experiments, but not in time). Next, to evaluate the effect of time in each specific experiment, we used the selected gene list and fitted a generalized linear model with a negative binomial distribution followed by LRT for each experiment separately. Genes with $p < 0.05$ (corrected) and fold change > 1.2 were considered significant. List of the detected genes, normalized counts, and p-values (FDR corrected) is presented in *Supplementary file 4*.

## Gene annotation and functional analysis
KEGG pathway analysis was performed using the 'SPIA' package (Signaling Pathway Impact Analysis) in R. Pathways with $p < 0.05$ and at least eight differentially expressed genes were considered significant. GO term enrichment analysis was performed using the 'clusterProfiler' package in R. Molecular function (MF) sub-ontologies were included in the analysis. The results of the inducible transcription analysis ($p < 0.05$, FDR corrected) are included in *Supplementary file 5* (complete list of enriched GO terms and genes) and in *Figure 1H* (representative GO term list). In the analysis of baseline transcription, we perform a second step of clustering in order to remove redundancy and identify global patterns across structures. After selecting the significantly enriched GO terms ($p < 0.05$, FDR corrected), we grouped together all GO terms that shared at least 50% identity of the differentially expressed genes in any of the structures (*Supplementary file 3*). As described in the Results section, few clusters were selected for presentation, and the expression levels of genes included in these clusters – across all time points and all structures – are presented as a heatmap in *Figure 1C*, *Figure 1—figure supplement 4*.

## smFISH analysis
For the IEG probes, selection for 'robust-expressing' cells was done as follows: We used the cocaine-naïve control data and after removing the non-expressing cells (cells expressing 0–1 puncta), the remaining cells were binned equally into three groups based on the per-cell expression levels, and the top 33% cells were defined 'robust expressors' or 'suprathreshold cells'. Thus, cells qualified as 'robust expressors' for a given IEG if they expressed at least the following number of puncta per

cell: *Arc* – 11, *Egr2* – 6, *Nr4a1* – 12, *Fos* – 5. For *Drd1* and *Drd2* expression, a threshold of 8 puncta/cell was implemented (*Gonzales et al., 2020*).

In order to identify the area with the highest density of IEG expressing cells in the striatum, we performed two-dimensional kernel density estimation using the function 'geom_density_2d' in R as in *Gonzales et al., 2020*. This function estimates two-dimensional kernel density with an axis-aligned bivariate normal kernel, evaluated on a square grid, while displaying the result with contours. The regions of highest density, within which at least 20% of the cells are found, were selected. This process was performed independently for each one of the replicas and the selected contours plotted. A list of the samples and number of cells included in the analysis is found in *Supplementary file 7*. Details of statistical analysis and results for smFISH data are summarized in *Supplementary file 6*. Raw data (puncta per cell) is available on Mendeley Data (http://dx.doi.org/10.17632/p5tsv2wpmg.1).

### Re-used data

Image reproduction: In the current study, we perform a comparison of the expression patterns and spatial distribution of IEGs following behavioral sensitization to cocaine. To this end, we compare the response to repeated and challenge cocaine exposures (novel data) to the response to acute cocaine, which was previously published (*Gonzales et al., 2020*). The reproduced images are the panels labeled 'acute' in *Figures 2B–D* and *3A–C*.

Data re-analysis: smFISH data presented in the manuscript relating to acute cocaine were previously published (*Gonzales et al., 2020*) and are included in the current manuscript for the sake of comparison to repeated and challenge cocaine (relevant to *Figures 2* and *3*, *Figure 2—figure supplement 2–S1, S2*, *Figure 3—figure supplement 1*). The reproduction of the data was approved by the editorial office of PNAS.

The locomotor sensitization data presented in *Figure 1B* is a summed representation of all mice collected for RNA-seq and smFISH analysis. Samples included in the RNA-seq analysis (n = 48) are derived from a subset of the mice (n = 71) analyzed by qPCR in *Mukherjee et al., 2018*, DOI: 10.7554/eLife.31220, while brain sections utilized for smFISH analysis were from mice that were also used for smFISH analysis in *Gonzales et al., 2020*; *Terem et al., 2020*.

## Acknowledgements

The authors appreciate the helpful critical comments of members of the Citri lab and Prof. Inbal Goshen on data, writing, and presentation. Prof. Ido Amit generously provided instruction and guidance on RNA-seq library preparation. Work in the Citri laboratory is funded by the European Research Council (ERC 770951), The Israel Science Foundation (1062/18, 393/12, 1796/12, and 2341/15), The Israel Anti-Drug Administration, EU Marie Curie (PCIG13-GA-2013–618201), the National Institute for Psychobiology in Israel, Hebrew University of Jerusalem Israel founded by the Charles E Smith family (109-15-16), an Adelis Award for Advances in Neuroscience, the Brain and Behavior Foundation (NARSAD 18795), German–Israel Foundation (2299–2291.1/2011), and Binational Israel–United States Foundation (2011266), the Milton Rosenbaum Endowment Fund for Research in Psychiatry, a seed grant from the Eric Roland Fund for interdisciplinary research administered by the ELSC, contributions from anonymous philanthropists in Los Angeles and Mexico City, as well as research support from the Safra Center for Brain Sciences (ELSC) and the Canadian Institute for Advanced Research (CIFAR). DM was funded by a 'Golden Opportunity Doctoral fellowship', as well as a 'Bridging' Post-Doctoral Fellowship from the Jerusalem Brain Community.

## Additional information

### Funding

| Funder | Grant reference number | Author |
| --- | --- | --- |
| Israel Science Foundation | 1062/18 | Ami Citri |
| European Research Council | ERC 770951 | Ami Citri |
| Israel Science Foundation | 393/12 | Ami Citri |

| Israel Science Foundation | 1796/12 | Ami Citri |
|---|---|---|
| Israel Science Foundation | 2341/15 | Ami Citri |
| The Israel Anti-Drug Administration | | Ami Citri |
| EU Marie Curie | PCIG13-GA-2013-618201 | Ami Citri |
| National Institute for Psychobiology in Israel, Hebrew University of Jerusalem | 109-15-16 | Ami Citri |
| Adelis Award for Advances in Neuroscience | | Ami Citri |
| Brain and Behavior Research Foundation | 18795 | Ami Citri |
| German-Israeli Foundation for Scientific Research and Development | 2299-2291.1/2011 | Ami Citri |
| US-Isral Binational Science Foundation | 2011266 | Ami Citri |
| The Milton Rosenbaum Endowment Fund for Research in Psychiatry | | Ami Citri |
| Prusiner-Abramsky Research Award in Clinical and Basic Neuroscience | | Ami Citri |
| Jerusalem Brain Community | JBC Gold PhD Scholarship | Diptendu Mukherjee |
| Jerusalem Brain Community | JBC Bridging Postdoctoral Scholarship | Diptendu Mukherjee |

The funders had no role in study design, data collection and interpretation, or the decision to submit the work for publication.

## Author contributions
Diptendu Mukherjee, Ben Jerry Gonzales, Conceptualization, Formal analysis, Investigation, Visualization, Methodology, Writing - original draft, Writing - review and editing; Reut Ashwal-Fluss, Data curation, Software, Formal analysis, Investigation, Visualization, Methodology, Writing - review and editing; Hagit Turm, Investigation, Methodology, Project administration; Maya Groysman, Resources; Ami Citri, Conceptualization, Resources, Supervision, Funding acquisition, Visualization, Methodology, Writing - original draft, Project administration, Writing - review and editing

## Author ORCIDs
Diptendu Mukherjee  https://orcid.org/0000-0002-9752-1026
Ben Jerry Gonzales  https://orcid.org/0000-0002-7011-4631
Ami Citri  https://orcid.org/0000-0002-9914-0278

## Ethics
Animal experimentation: All animal protocols (# NS-13-13660-3; NS-13-13895-3; NS-15-14326-3; NS-16-14644-2; NS-14667-3; NS-16-14856-3; NS-19-15753-3) were approved by the Institutional Animal Care and Use Committees at the Hebrew University of Jerusalem and were in accordance with the National Institutes of Health Guide for the Care and Use of Laboratory Animals.

## Decision letter and Author response
Decision letter https://doi.org/10.7554/eLife.65228.sa1
Author response https://doi.org/10.7554/eLife.65228.sa2

# Additional files

## Supplementary files

- Supplementary file 1. Distribution of sequencing libraries analyzed in this study.
- Supplementary file 2. Normalized reads of baseline shifted genes (tabs corresponding to individual structures).
- Supplementary file 3. Clusters of Gene Ontology (GO) analysis and corresponding genes.
- Supplementary file 4. Normalized reads of induced genes (tabs correspond to structures).
- Supplementary file 5. Gene Ontology annotation of cocaine-induced genes in the dorsal striatum.
- Supplementary file 6. Statistical analysis.
- Supplementary file 7. Distribution of cell numbers among replicates in smFISH analysis.
- Transparent reporting form

## Data availability

Source data file for RNA-seq and smFISH experiments are available at NCBI GEO: GSE158588, and https://doi.org/10.17632/p5tsv2wpmg.1.

The following datasets were generated:

| Author(s) | Year | Dataset title | Dataset URL | Database and Identifier |
|---|---|---|---|---|
| Mukherjee D, Gonzales BJ, Ashwal-Fluss R, Turm H, Groysman M, Citri A | 2021 | RNA-seq of five brain structures after repeated exposure to cocaine | http://www.ncbi.nlm.nih.gov/geo/query/acc.cgi?acc=GSE158588 | NCBI Gene Expression Omnibus, GSE158588 |
| Mukherjee D, Gonzales BJ, Ashwal-Fluss R, Turm H, Groysman M, Citri A | 2021 | smFISH data of IEG expression in the dorsal striatum after acute, repeated, and challenge cocaine exposures | https://data.mendeley.com/datasets/p5tsv2wpmg/draft?a=ab883129-19c3-40e0-8431-906929984809 | Mendeley Data, 10.17632/p5tsv2wpmg.1 |

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
