## [Decision Letter]

**Acceptance summary:**

Drugs of abuse like cocaine alter gene expression patterns in brain reward circuits, and this transcriptional response is essential for drug-induced cellular and behavioral plasticity. Supported by careful transcriptional profiling, single-molecule RNA analyses, and genetic perturbations, this manuscript identifies Egr2 as a top marker of cocaine-activated neuronal ensembles in the dorsal striatum and establishes a role for this gene in cocaine response. This work will be of broad interest to researchers studying programmed gene responses, and also to the addiction neuroscience community.

**Decision letter after peer review:**

Thank you for submitting your article "Egr2 induction in Drd1^+^ ensembles of the ventrolateral striatum supports the development of cocaine reward" for consideration by *eLife*. Your article has been reviewed by three peer reviewers, one of whom is a member of our Board of Reviewing Editors, and the evaluation has been overseen by Kate Wassum as the Senior Editor. The reviewers have opted to remain anonymous.

The reviewers have discussed the reviews with one another, and the Reviewing Editor has drafted this decision to help you prepare a revised submission.

As the editors have judged that your manuscript is of interest, but as described below that additional experiments are required before it is published, we would like to draw your attention to changes in our revision policy that we have made in response to COVID-19 (https://elifesciences.org/articles/57162). First, because many researchers have temporarily lost access to the labs, we will give authors as much time as they need to submit revised manuscripts. We are also offering, if you choose, to post the manuscript to bioRxiv (if it is not already there) along with this decision letter and a formal designation that the manuscript is "in revision at *eLife*". Please let us know if you would like to pursue this option.

Summary:

Drugs of abuse like cocaine alter gene expression patterns in brain reward circuits, and this transcriptional response is essential for drug-induced cellular and behavioral plasticity. In this manuscript, the authors use careful transcriptional profiling, single-molecule RNA analyses, and genetic perturbations to further define the cellular populations and transcriptional response programs that contribute to cocaine response. Together, the results of this manuscript establish that cocaine activates an Egr2^+^ neuronal ensemble in the ventrolateral striatum and reveal a role for Egr2 in cocaine-related behavioral responses. Overall, the results of the manuscript are compellingly presented, and validation of the role for Egr2^+^ ensembles with distinct approaches supports the conclusions of the manuscript. A limitation of this work is that it does not define the potential gene targets of Egr2 and does not address how Egr2-regulated gene programs contribute to VLS function and physiology. Further, it does not demonstrate the sufficiency of Egr2 for cocaine-related behavioral plasticity. However, this is a solid manuscript that makes an important and detailed contribution to our understanding of addiction-relevant transcriptional regulatory mechanisms. This work will be of broad interest to researchers studying programmed gene responses, and also to the addiction neuroscience community.

Essential revisions:

1) The results presented in Figure 1I are intriguing and suggest that inhibition of Egr2^+^ neurons in the dorsal striatum blocks development of cocaine locomotor sensitization. However, this result is based on a small sample size (n=3 per group), and is missing some key controls (most notably, delivery of DIO-Kir2.1 to Cre- animals). Further, the authors do not show any validation that this approach resulted in silencing of Egr2^+^ DS neurons. At the very least, repeating the Kir2.1 overexpression experiment and validating the approach would significantly strengthen the author's conclusions.

2) The authors should provide the scoring or stereotypies that usually developed following repeated exposure (20 mg/kg) to cocaine. This is important because it could be simply that mice biased their behavioral responses towards stereotypies. This issue should be clarified. Similarly, in the discussion the authors mention that VLS receives strong inputs from cortical areas conveying information arising from limbs and mouth, which are both highly impacted following repeated exposure to cocaine (stereotypies). The authors had therefore a unique opportunity to directly test whether neuronal ensembles within this striatal domain could be involved in the development of motor stereotypies induced by psychostimulants. The authors should seriously consider this option.

3) Part of the data of this study has already been used in a recent publication of the lab in Current Biology (Gonzales et al., 2020). While this is not a problem as there are different cocaine programs in this study, this point should be clearly mentioned. Similarly, Figure 1B, acute of this manuscript contains the same images as Figure 1C of the current biology: these images should be either changed, or “from Gonzales et al., 2020” should be in the legend of the Figure. The same applies to Figure 3A, Drd1, acute. The overlap of the data of Figure 1A-C in Mukherjee et al., 2018 and this study should also be clarified in the text and figures.

4) Figure 3 panel E and G: In contrast to what they state, these experiments do not support direct evidence that Egr2 in VLS D1 neurons contribute to the rewarding properties induced by cocaine. Single values from panel E clearly show that only 1 out of 6 mice do not display CPP following DREADD activation. The decreased CPP therefore relies only on this point. Repeating this experiment would strengthen the claims that the authors make. Moreover, the dose of CNO used is particularly high. The experiments should be performed with a lower dose of CNO and include validation showing that Egr2 neurons expressing hM4Di are indeed inactivated.

5) Similarly, in their last experiment, the authors provide elegant loss of function experiments by using an inhibitory DREADD (hM4Di) and an Egr2 mutant which does not bind to DNA, to demonstrate the implication of the Egr2 pathway in the induction of cocaine CPP. However, the authors mention it is necessary for cocaine CPP, while all hM4Di mice except one, and all DNEgr2 except 2, still exhibited CPP after the manipulation. The authors need to temper their conclusion to reflect the data and say they demonstrated an implication of Egr2 in the VLS in cocaine CPP. Necessity would be reflected by an absence of CPP. In the future, the authors might consider bilateral injections to obtain stronger behavioral effects.

6) Please ensure full statistical reporting in the main manuscript (e.g., test statistic, degrees of freedom, in addition to p value).

[Editors' note: further revisions were suggested prior to acceptance, as described below.]

Thank you for resubmitting your work entitled "Egr2 induction in Drd1^+^ SPNs of the ventrolateral striatum supports cocaine place preference in mice" for further consideration by *eLife*. Your revised article has been evaluated by Kate Wassum (Senior Editor) and a Reviewing Editor.

Summary:

Drugs of abuse like cocaine alter gene expression patterns in brain reward circuits, and this transcriptional response is essential for drug-induced cellular and behavioral plasticity. Supported by careful transcriptional profiling, single-molecule RNA analyses, and genetic perturbations, this manuscript identifies Egr2 as a top marker of cocaine-activated neuronal ensembles in the dorsal striatum and establishes a role for this gene in cocaine response. This work will be of broad interest to researchers studying programmed gene responses, and also to the addiction neuroscience community.

The manuscript has been improved but there are some remaining issues that need to be addressed before acceptance, as outlined below:

1) Please adjust the title to more accurately reflect the finding of the manuscript. Specifically, we suggest replacing "supports" with "contributes", to make it more clear that Egr2 neuron silencing in SPNs of the VLS is not abolishing cocaine CPP, but rather dampening it. Moreover, it may not be accurate to mention Drd1^+^ SPNs, as a significant proportion of Egr2^+^ neurons are Drd2^+^ (about 41.6% if including Egr2^+^Drd2^+^ and Egr2^+^Drd1^+^Drd2^+^). While there are more Drd1^+^ (about 73.6% if including Egr2^+^Drd1^+^ and Egr2^+^Drd1^+^Drd2^+^), this is still less than twice more. We suggest the following title: "Egr2 induction in SPNs of the ventrolateral striatum contributes to cocaine place preference in mice".

2) The new results included in Figure 3—figure supplement 3 supports the role of Egr2 neurons in the context of cocaine CPP and uses a lower dose of CNO. This experiment addresses several major concerns raised in the prior round of review and is as or more convincing than the current Figure 3F. We suggest including the data from Figure 3—figure supplement 3 in the main part of the paper and not as a supplemental figure.

---

## [Author Response]

Essential revisions:1) The results presented in Figure 1I are intriguing and suggest that inhibition of Egr2^+^ neurons in the dorsal striatum blocks development of cocaine locomotor sensitization. However, this result is based on a small sample size (n=3 per group), and is missing some key controls (most notably, delivery of DIO-Kir2.1 to Cre- animals). Further, the authors do not show any validation that this approach resulted in silencing of Egr2^+^ DS neurons. At the very least, repeating the Kir2.1 overexpression experiment and validating the approach would significantly strengthen the author's conclusions.

We accept the reviewers’ critique of this experiment. Currently we are limited in our capacity to develop additional experiments. We believe that the results of this experiment are valid and have applied a similar approach (of Kir2.1 inhibition of Egr2^+^-expressing neurons) in previous studies (Atlan et al., 2018; Terem et al., 2020), in which we characterized the physiological impact of Kir2.1 expression (Atlan et al., 2018; Figure S2B). However, we accept the reviewers’ concern of the number of mice in the study is small. As this experiment is not a crucial building block of this manuscript, we have simply removed it from the revised version, so as not to cause an unwarranted delay in publication.

2) The authors should provide the scoring or stereotypies that usually developed following repeated exposure (20 mg/kg) to cocaine. This is important because it could be simply that mice biased their behavioral responses towards stereotypies. This issue should be clarified. Similarly, in the discussion the authors mention that VLS receives strong inputs from cortical areas conveying information arising from limbs and mouth, which are both highly impacted following repeated exposure to cocaine (stereotypies). The authors had therefore a unique opportunity to directly test whether neuronal ensembles within this striatal domain could be involved in the development of motor stereotypies induced by psychostimulants. The authors should seriously consider this option.

This is truly a fascinating notion and is the basis of work we are currently developing in the lab, addressing the role of VLS neurons in cocaine stereotypy. Since the focus of the experiments described in the current manuscript were on locomotor sensitization and place preference, the paradigm and behavior tracking protocol were tailored to address these questions, rather than address the development of stereotypies. Our understanding, based on the literature and our cumulative experience, is that the behavioral output induced by cocaine depends on the concentration of the drug applied and the context (size of the arena) in which the drug is administered. Therefore, we applied 10 mg/kg for CPP experiments and 20 mg/kg for locomotor sensitization. Orofacial stereotypies are reputed to emerge as the major behavioral output at higher concentrations of cocaine (30 mg/kg) under conditions in which the path of the mice is restricted, by placing them in smaller arenas (Xu et al., 1994; Blanchard, 2000; Caster and Kuhn, 2009; Giros et al., 1996).

In any case, we are not able to quantify stereotypies in the experiments included in the manuscript, as these experiments were documented solely with a top-view camera with the objective of quantifying locomotion.

3) Part of the data of this study has already been used in a recent publication of the lab in PNAS (Gonzales et al., 2020). While this is not a problem as there are different cocaine programs in this study, this point should be clearly mentioned. Similarly, Figure 1B, acute of this manuscript contains the same images as Figure 1C of the current biology: these images should be either changed, or 'from Gonzales et al. 2020' should be in the legend of the Figure. The same applies to Figure 3A, Drd1, acute. The overlap of the data of Figure 1A-C in Mukherjee et al., 2018 and this study should also be clarified in the text and figures.

We acknowledged the reuse of data in the original submission and following the reviewer’s suggestions have added a subsection “Re-used data”. This section details all overlap with data used in other manuscripts:

Regarding specific comments made by the reviewer – the panel in Figure 1B is a summary of the behavior of all mice included in the current study (partially overlapping with mice included in previous studies, as defined in the Materials and methods), and has not been previously published. With regard to the request to relate in the figure legends to panels that appeared in previous publications, we have a reference, as requested, in the legend of Figure 2 and Figure 3: “Images relating to acute cocaine (in panels B, C and D) were replicated from Gonzales et al., 2020, PNAS, with permission).”

4) Figure 3 panel E and G: In contrast to what they state, these experiments do not support direct evidence that Egr2 in VLS D1 neurons contribute to the rewarding properties induced by cocaine. Single values from panel E clearly show that only 1 out of 6 mice do not display CPP following DREADD activation. The decreased CPP therefore relies only on this point. Repeating this experiment would strengthen the claims that the authors make. Moreover, the dose of CNO used is particularly high. The experiments should be performed with a lower dose of CNO and include validation showing that Egr2 neurons expressing hM4Di are indeed inactivated.

We thank the reviewers for this comment. We have indeed, in the past, performed this experiment in additional iterations, providing confidence in the validity of our observations. We have now included an additional experiment as Figure 3—figure supplement 3. This experiment was based on a slightly different CPP paradigm, in which the preference of mice was tested repeatedly, interleaved between each of 3 conditioning sessions, in contrast to the experiment included in Figure 3E-F, in which we tested the preference of mice only once, following 3 consecutive conditioning days. The interleaved protocol supports analysis of behavior during the induction of *Egr2* & *CRE* expression (following the first cocaine exposure), gradually leading up to hM4Di recombination and behavioral impact. Furthermore, in this experiment, controls were hM4Di expressing mice which were exposed to saline, while in the experiment included in Figure 3E-F, control mice (expressing mCherry) were exposed to CNO, similar to experimental mice (expressing hM4Di). Finally, in the experiment included in Figure 3—figure supplement 3, CNO was administered at 5 mg/kg.

The relevant section of text in the manuscript currently reads: “The selective enrichment of Egr2 induction within VLS Drd1^+^ neurons suggests a causal role for this neuronal population in supporting cocaine-conditioned behaviors. […] We therefore conclude that VLS Egr2^+^-expressing neurons contribute to the development of cocaine-seeking behavior, with no obvious impact on locomotor aspects of cocaine-driven behavior.”

With regard to further validation of the action of CNO on hM4Di DREADDs, we have previously performed an electrophysiological evaluation of the impact of CNO-hM4Di on excitability of Egr2^+^ neurons (albeit claustrum Egr2^+^ neurons; Atlan et al., 2018).

**Author response image 1. sa2fig1:** Efficacy and efficiency of transduction of hM4Di infection of CL_Egr2+_ neurons. (A) Representative traces of whole-cell current clamp recordings from CL_Egr2+_ neurons expressing the hM4Di DREADD, in the presence or absence of CNO (clozapine-N-oxide; 1 μM). 50-250 pA current injections (right) demonstrate the reduction of excitability in cells expressing hM4Di before(middle) and after (right) application of CNO. (B) Summary graph of the impact of CNO on the excitability of neurons expressing hM4Di.

While we do not have a direct measurement in VLS Egr2^+^ neurons and are limited in our capacity to perform these experiments currently, we have observed, in unpublished immunostaining experiments, that exposure to CNO inhibited the expression of Fos in hM4Di-expressing Egr2^+^ neurons in the claustrum (see Author response image 2). As we are utilizing the same reagents (mice, viruses and CNO) in the current study, we assume the effect to be similar.

**Author response image 2. sa2fig2:** hM4Di-CNO inhibits cocaine-induced Fos induction in Egr2^+^ claustral neurons. The claustrum of Egr2-CRE mice was transduced with AAV viruses conditionally expressing hM4Di. 3 weeks later, following habituation to ip saline injections, mice were injected (i.p. 10mg/kg) with CNO, and 30 minutes later with cocaine (20mg/kg), sacrificed 1.5 hrs later. Sections were submitted to immunostaining for Fos (magenta). The representative overview image (left) and merged image (right) demonstrate the absence of co-localization of Fos staining in cells expressing the hM4Di DREADD.

Furthermore, a similar experiment (CNO action on hM4Di prior to cocaine), performed in the VLS of Drd1-Cre mice (in a separate study) provided essentially the same results, increasing our confidence that ligation of hM4Di by CNO acts to inhibit VLS neurons (see Author response image 3).

**Author response image 3. sa2fig3:** hM4Di-CNO inhibits cocaine-induced Fos induction in VLS Drd1^+^-neurons. The VLS of Drd1^+^-CRE mice was transduced with AAV viruses conditionally expressing hM4Di (red). Following habituation to ip saline injections, mice were injected with CNO (i.p. 10 mg/kg), and 30 minutes later with cocaine, sacrificed 1.5 hrs later. Sections were immunostained for Fos (green). A representative image from a mouse injected with saline before cocaine (left) demonstrates co-localization of green+red cells (white arrows), which are not observed in the representative image (right) from the mouse exposed to CNO prior to cocaine.

5) Similarly, in their last experiment, the authors provide elegant loss of function experiments by using an inhibitory DREADD (hM4Di) and an Egr2 mutant which does not bind to DNA, to demonstrate the implication of the Egr2 pathway in the induction of cocaine CPP. However, the authors mention it is necessary for cocaine CPP, while all hM4Di mice except one, and all DNEgr2 except 2, still exhibited CPP after the manipulation. The authors need to temper their conclusion to reflect the data and say they demonstrated an implication of Egr2 in the VLS in cocaine CPP. Necessity would be reflected by an absence of CPP. In the future, the authors might consider bilateral injections to obtain stronger behavioral effects.

We hope the additional experiment, now included in Figure 3—figure supplement 3 appeases the reviewers’ concerns regarding the reliability of the effect of hM4Di DREADD inhibition of VLS Egr2^+^ neurons. All experiments were performed with bilateral injections, which we now state more clearly in the text (single hemispheres are shown in the figure only for illustration). Following the reviewers’ suggestion, we have tempered the description of the results and conclusions in the text, and the title of this section has also been revised to read “Implication of VLS Egr2 transcriptional activity in the development of cocaine-seeking behavior”.

6) Please ensure full statistical reporting in the main manuscript (e.g., test statistic, degrees of freedom, in addition to p value).

We have made sure to add full statistical reporting to the main manuscript, in addition to the Supplementary file 6, which provides a comprehensive description of the statistical methods and results.

[Editors' note: further revisions were suggested prior to acceptance, as described below.]

The manuscript has been improved but there are some remaining issues that need to be addressed before acceptance, as outlined below:1) Please adjust the title to more accurately reflect the finding of the manuscript. Specifically, we suggest replacing "supports" with "contributes", to make it more clear that Egr2 neuron silencing in SPNs of the VLS is not abolishing cocaine CPP, but rather dampening it. Moreover, it may not be accurate to mention Drd1^+^ SPNs, as a significant proportion of Egr2^+^ neurons are Drd2^+^ (about 41.6% if including Egr2^+^Drd2^+^ and Egr2^+^Drd1^+^Drd2^+^). While there are more Drd1^+^ (about 73.6% if including Egr2^+^Drd1^+^ and Egr2^+^Drd1^+^Drd2^+^), this is still less than twice more. We suggest the following title: "Egr2 induction in SPNs of the ventrolateral striatum contributes to cocaine place preference in mice".

As requested, we have revised the title of the manuscript, which now reads: “Egr2 induction in spiny projection neurons of the ventrolateral striatum contributes to cocaine place preference in mice”.

2) The new results included in Figure 3—figure supplement 3 supports the role of Egr2 neurons in the context of cocaine CPP and uses a lower dose of CNO. This experiment addresses several major concerns raised in the prior round of review and is as or more convincing than the current Figure 3F. We suggest including the data from Figure 3—figure supplement 3 in the main part of the paper and not as a supplemental figure.

As further requested, we have transferred the experiment included in Figure 3—figure supplement 3 into the main figure, and it now comprises panels D, E of Figure 3.